# Exploration is associated with socioeconomic disparities in learning and academic achievement in adolescence

Alexandra L. Decker [1,2] ✉, Julia Leonard [3,12], Rachel Romeo[4,12], Joseph Itiat[1], Nicholas A. Hubbard [5], Clemens C. C. Bauer[1,6,7], Hannah Grotzinger[8], Melissa A. Giebler[9], Yesi Camacho Torres[1,10], Andrea Imhof[11] & John D. E. Gabrieli[1]

Adolescents from lower socioeconomic status backgrounds often underperform on tests of learning and academic achievement. Existing theories propose that these disparities reflect not only external constraints, like limited resources, but also internal decision strategies that adapt to the early environment and influence learning. These theories predict that adolescents from lower socioeconomic status backgrounds explore less and exploit more, which, in turn, reduces learning and academic achievement. Here, we test this possibility and show that lower socioeconomic status in adolescence is associated with less exploration on a reward learning task ($n = 124$, 12–14-year-olds from the United States). Computational modeling revealed that reduced exploration was related to higher loss aversion. Reduced exploration also mediated socioeconomic differences in task performance, school grades, and, in a lower-socioeconomic status subsample, academic skills. These findings raise the possibility that learning disparities across socioeconomic status relate not only to external constraints but also to internal decision strategies and provide some mechanistic insight into the academic achievement gap.

Children from lower socioeconomic status (SES) backgrounds lag behind their higher-SES peers in learning and academic achievement by up to several years on average[1–5]. These disparities are linked to external environmental constraints, including more limited access to educational and financial resources[6–10]. However, emerging theories propose that early economic adversity might also shape internal decision strategies that in turn, influence learning and academic achievement[11–18]. One theory suggests that those from lower-SES backgrounds might rely more on exploitation to secure immediate resources rather than exploration, which facilitates broad information gathering and learning[11–18]. Although reduced exploration could theoretically contribute to learning and academic achievement gaps[11], direct empirical evidence linking SES, exploratory behavior, learning, and academic achievement is lacking. Here, we address this gap by examining associations between SES, exploratory decision-making, and measures of learning and academic achievement, including task performance, school grades, and standardized academic skills.

[1]McGovern Institute for Brain Research, Massachusetts Institute of Technology, Cambridge, MA, USA. [2]Department of Psychological and Brain Sciences, Washington University in St. Louis, St. Louis, MO, USA. [3]Department of Psychology, Yale University, New Haven, CT, USA. [4]Departments of Human Development & Quantitative Methodology and Hearing & Speech Sciences, and Program in Neuroscience & Cognitive Science, University of Maryland College Park, Baltimore, MD, USA. [5]Center for Brain, Biology, and Behavior, University of Nebraska, Lincoln, NE, USA. [6]Department of Psychology, Northeastern University, Boston, MA, USA. [7]Center for Precision Psychiatry, Massachusetts General Hospital, Boston, MA, USA. [8]Department of Psychological & Brain Sciences, University of California, Santa Barbara, CA, USA. [9]Teachers College, Columbia University, New York, NY, USA. [10]Luskin School of Public Affairs, University of California, Los Angeles, CA, USA. [11]Department of Psychology, University of Oregon, Eugene, OR, USA. [12]These authors contributed equally: Julia Leonard, Rachel Romeo. ✉ e-mail: adecker@mit.edu

Balancing exploration and exploitation involves a trade-off[19,20] that has consequences for learning. Exploring in search of new information facilitates learning through experience-driven feedback[21,22]. New knowledge can then be used to update models of the world and optimize future decisions. Exploration, however, carries costs: In the short term, exploring demands time and resources and often leads to suboptimal immediate outcomes[19,20]. For example, learning a new skill requires a substantial time investment and leads to short-term failures before one can enjoy the rewards of mastery. Exploitation, by contrast, maximizes immediate reward by leveraging existing knowledge but can constrain new learning. This tension between immediate returns and long-term knowledge acquisition may shape how individuals navigate and learn from their environment.

The human species is distinguished by an unusually extended childhood—a feature hypothesized to have evolved to enable extensive exploration before the responsibilities of adulthood take hold[21–40]. This extended period gives children ample time to gather information, learn the structure of their environment, and optimize their decision strategies accordingly[23]. Information learned during this exploratorty period can then be used to optimize decision-making later in life[21–26,30–32,34,36,39,40]. Consistent with this idea, children and adolescents have been shown to be more exploratory than adults[23–27,29,33,35,38,39,41] (though see other studies that did not report this pattern[32,42]). Children have been shown to make more exploratory choices on decision tasks than adults, which boosts learning of task structure, allowing children to adapt better to changes in task rules[27–29,40]. Adolescents also exhibit more risky exploratory behavior than adults[22,24,25,30,33,35,38], which is linked to superior performance on foraging tasks[25]. Based on existing frameworks[21–40], if this exploratory period is missed, opportunities for broad learning would be constrained, leading to less optimal decisions later.

However, it has been hypothesized that children and adolescents from resource-constrained or unpredictable environments might adapt by shifting decision strategies in favor of exploitation over exploration[11–14,16–18,43–45]. This shift is proposed to be adaptive. When resources are scarce or unpredictable, exploiting helps ensure basic needs are met[11–14,16–18,43]. Exploring, by contrast, can lead to short-term costs that may be difficult to overcome with limited resources. While research linking SES to exploration is limited, children[43,44], adolescents[43], and adults[18] who experienced higher levels of adversity[18,43] and environmental unpredictability[44] (more prevalent in lower-SES environments[46,47]) explored less on foraging[18,44] and reward decision tasks[43,44]. Furthermore, adolescents[48] and adults[14] who were from lower-SES backgrounds in childhood tended to exploit by settling for smaller, certain, or immediate rewards over larger, uncertain, or delayed ones. Non-human animal studies complement this cross-sectional human research, showing that resource scarcity (e.g., food deprivation)[45,49,50] causally reduces exploration later in life.

While adaptive for meeting basic needs, reduced exploration could limit the novelty and feedback that drives learning and academic achievement[11,12,43], especially when generalized across settings. At present, however, whether decision strategies relate to gaps in learning and academic achievement across SES is unclear. This is because decision-making and learning across SES have been studied in separate literatures. Empirically examining links between decision strategies and learning or achievement outcomes could offer insight into how these strategies may be associated with learning disparities across SES.

Furthermore, identifying factors that might underlie greater exploitation in lower-SES adolescents could inform strength-based interventions. One contributing factor to reduced exploration could be that lower-SES adolescents weigh potential losses more heavily in decision making than equivalent gains (they are more loss-averse). This could be an adaptive response to having limited resources to compensate for losses (though, notably, some research indicates that

extreme poverty heightens risk-seeking due to having so little left to lose[51,52]). However, another (not mutually exclusive) possibility is that lower-SES adolescents explore less because they believe exploring will not lead to rewards–beliefs that could develop in resource-constrained environments with more limited opportunities for rewarding outcomes. This latter possibility aligns with research linking lower SES to pessimistic expectations about the future[53,54]. To date, whether these factors relate to SES-based differences in exploration remains unclear.

The current study examined the association between SES, exploratory behavior, task performance, and academic achievement, measured by grades and academic skills. We focused on 12–14-year-old children and adolescents, an age range associated with heightened exploration and learning[22,24,25,30–33,35,38]. We tested three a priori hypotheses. First, based on prior research[11,12,18,43–45,49,50], we assessed whether lower SES would correlate with less exploration on a reward decision task. Second, we tested whether reduced exploration would be associated with higher loss aversion or reduced initial expectations that exploring would lead to positive outcomes. Third, based on literature linking exploration to learning[12,21–24,26,29,30], we tested whether SES-based differences in exploration were related to task performance, grades, and academic skills. The second hypothesis was conceived before data analysis but after the study was designed. This study was not pre-registered.

Adolescents from diverse socioeconomic backgrounds ($n = 124$; household income range: $2,000–1.25 million annually; Fig. 1a, b) completed the Balloon Emotional Learning Task (BELT; Fig. 1c)[43,55,56]. On each trial, adolescents inflated a virtual balloon to earn points[43,55,56]. Each time participants pumped the balloon, they earned 1 point, and the balloon grew in size. However, pumping too much risked the balloon exploding and the loss of all points for a trial. Participants could, therefore, stop pumping at any point to secure points for a trial. However, choosing to stop pumping forgoed opportunities for more points and information about the balloon's popping threshold. The task, therefore, required adolescents to balance securing points (exploiting) with earning more points and information about the balloon's explosion limits (exploring).

The task featured three types of balloons, each with a unique explosion threshold. To maximize points, adolescents had to learn the balloon thresholds through trial and error. There was an orange balloon that exploded after 8 pumps and favored a more exploitative strategy of securing points earlier (the short balloon). There was a pink balloon that had a high explosion threshold–20 pumps–that favored an exploratory strategy (the long balloon). There was a blue balloon that had a variable explosion threshold–8, 14, or 20 pumps (the unreliable balloon). Through pumping and experiencing explosions, participants could learn these explosion thresholds and optimize decisions. Performance was assessed by points earned. Exploration was measured through the number of pumps and explosions, where more pumps and explosions indicated a greater willingness to explore the limits of the task conditions. Academic achievement was measured using parent-reported grades and a standardized test of academic skills (See Methods).

To preview, we found that lower-SES adolescents explored less than their higher-SES peers on the BELT. Higher loss aversion among lower-SES adolescents mediated SES-related differences in exploration. Furthermore, reduced exploration mediated SES-based differences in task performance, school grades, and, in a lower-SES subsample, academic skills. Exploratory behavior also fluctuated across the task in response to shifting reward outcomes, where greater rewards preceded more exploration, suggesting rewards boost exploration. These findings align with theoretical predictions that lower SES might shape internal decision strategies by reducing exploration, and in turn, constrain learning and academic achievement[11].

**a** Household Income Distribution

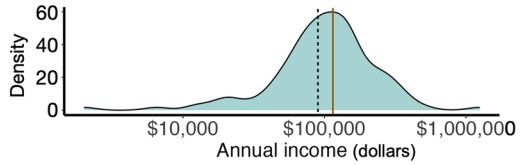

**b** Parental Education Distribution

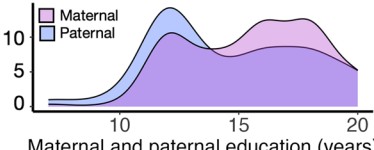

**c** Balloon Emotional Learning Task

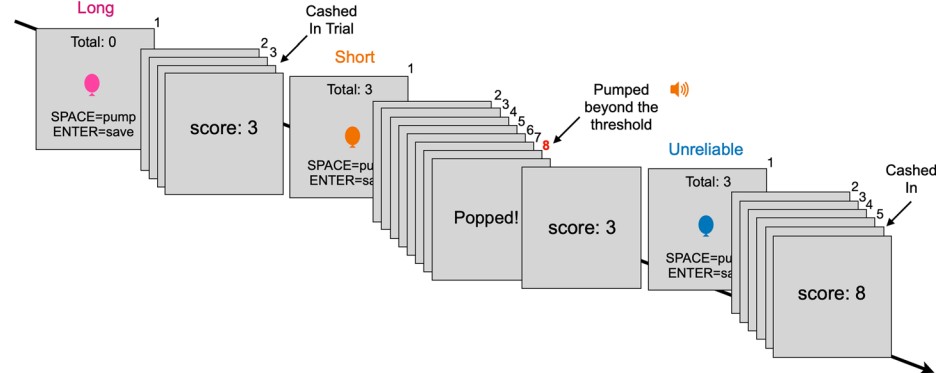

**Fig. 1 | Annual Household Income and Parental Education Distributions, and the Balloon Emotional Learning Task Schematic. a** The annual household income distribution of the sample plotted on a log scale (*n* = 121; median = $100,000; range: $2000–$1.25 million). The dashed line marks the low-income threshold for a family of four in 2019 in the Boston-Cambridge-Quincy Metro Area ($89,200), and the brown solid line marks the median household income in 2019 in the Boston-Cambridge-Quincy Metro Area ($113,300); retrieved from: https://www.huduser.gov/portal/datasets/il.html#2019_query. **b** The distribution of parental years of education separately for maternal/parent 1 (median: 16 years; range: 7–20;

*n* = 124, pink) and paternal/parent 2 education levels (median: 14 years; range: 7–20; *n* = 119, blue). In **a-b**, y-axes reflect density distributions as percentages. **c** On each trial, participants inflated a virtual balloon to earn points. With each pump, the balloon grew, and participants earned 1 point. Pumping too much led to a balloon explosion and losing all points for a trial. Participants could, therefore, stop pumping and save their points instead of continuing to pump for more points. There were three balloon types, each with a unique color (pink, orange, blue) and explosion threshold. To maximize points, adolescents had to learn the explosion limits through pumping and experiencing explosions.

## Results

### Adolescents incrementally learned across trials

Adolescents demonstrated signatures of learning in the BELT: their performance improved as they gained experience with the task, as shown by a positive association between trial number and points earned (z = 5.91, *p* < 0.001, *p*-adjusted < 0.001, b = 0.005, 95% CI [0.003, 0.006]). The interaction between trial number and balloon type was significant, indicating steeper incremental gains in points earned on long than unreliable balloon trials (z = 2.24, *p* = 0.025, *p*-adjusted = 0.030, b = 0.003, 95% CI [0.0004, 0.006]) and on short than unreliable balloon trials (z = 4.32, *p* < 0.001, *p*-adjusted <0.001, b = 0.008, 95% CI [0.004, 0.012]). Trial number was positively related to points on the long (z = 4.65, *p* < 0.001, *p*-adjusted <0.001, b = 0.005, 95% CI [0.002, 0.007]) and short balloon trials (z = 7.51, *p* < 0.001, *p*-adjusted <0.001, b = 0.009, 95% CI [0.007, 0.01]), but there was no statistically significant association between trial number and points earned on unreliable balloon trials (z = 1.27, *p* = 0.204, *p*-adjusted = 0.204, b = 0.001, 95% CI [−0.001, 0.003]). These findings suggest that adolescents incrementally learned the explosion thresholds of the long and short balloons, which had reliable popping thresholds. Analysis S1 reports further evidence of learning on long and short balloon trials.

### Exploratory pumping and explosions facilitated better task performance

To assess the role of exploration in learning, we tested whether individuals who experienced more pumps and explosions early in the task earned more points later (even after controlling for points earned early). Therefore, we tested whether more pumping and explosions early, in the first third of the task (during initial learning) predicted

more points earned later, in the last third of the task (representing the final expression of learning). Consistent with the idea that exploring preceded the adoption of more optimal strategies, we found that a higher mean number of pumps (t(122) = 3.03, *p* = 0.003, *p*-adjusted = 0.008, ρ = 0.266, 95% CI [0.090, 0.424]) and explosions (t(122) = 2.81, *p* = 0.006, *p*-adjusted = 0.009, ρ = 0.248, 95% CI [0.072, 0.409]) in the first third of the task predicted a higher mean number of points earned in the last third of the task. Early exploration therefore appeared to facilitate better performance later, implicating exploratory behavior–and explosion-related feedback–in the learning process.

### Loss aversion mediates socioeconomic disparities in task-based exploration

We next tested whether lower SES correlated with reduced exploration. We used pumping and explosions on unreliable balloon trials to index exploration because pumping and explosions on short and long balloon trials were confounded by learning effects (Supplementary Note 1), though results did not differ when using all balloon types (Supplementary Note 2). As hypothesized, lower SES was associated with fewer pumps (z = 3.63, *p* < .001, *p*-adjusted = 0.001, b = 0.079, 95% CI [0.036, 0.121]; Fig. 2a) and explosions (z = 3.25, *p* = 0.001, *p*-adjusted = 0.003, b = 0.26, 95% CI [0.102, 0.418]; Fig. 2b). Thus, lower SES was associated with less extensive sampling and less explosion-related feedback about the balloon's explosion limits.

But why might lower-SES adolescents explore less? Lower-SES adolescents may have been more loss-averse–preferring to limit losses by exploiting for guaranteed rewards. Alternatively, lower-SES adolescents might begin the task with more negative expectations, believing that the balloon will not yield as many rewards and be more likely to

### Socioeconomic Status Correlated With Exploration

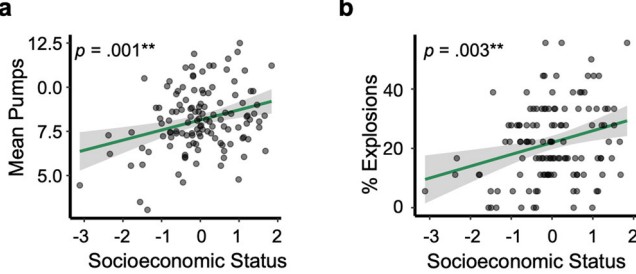

### Socioeconomic Status Correlated with Loss Aversion

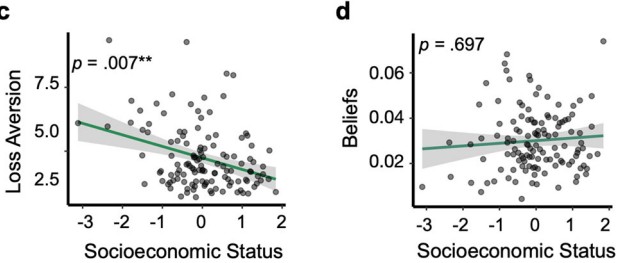

**Fig. 2 | Socioeconomic status correlated with exploration and loss aversion (n = 124). a** Higher socioeconomic status (SES) correlated with more pumping. Figure **a** shows this relationship for unreliable balloon trials but results are unchanged when collapsing across all trial types. Statistical test: Mixed-effects negative binomial regression (z = 3.63, p < 0.001, p-adjusted = 0.001, b = 0.079, 95% CI [0.036, 0.121]). **b** Higher SES correlated with more balloon explosions on unreliable balloon trials. Figure b shows this relationship for unreliable balloon trials but results are unchanged when collapsing across all trial types. Statistical test: Mixed-effects logistic regression (z = 3.25, p = 0.001, p-adjusted = 0.003, b = 0.26, 95% CI [0.102, 0.418]). **c** Lower SES correlated with higher loss aversion. Statistical test: Spearman's rank-order correlation (t(122) = −2.95, p = 0.004, p-adjusted = 0.007, ρ = −0.26, 95% CI [−0.42, −0.08]). **d** There was no statistically significant association between SES and initial beliefs about the likelihood that the balloon would explode when pumped. Statistical test: Spearman's rank-order correlation (t(122) = 0.432, p = 0.667, p-adjusted = 0.697, ρ = 0.04, 95% CI [−0.14, 0.22]). In **a** & **b**, points reflect participant-level means and percentages for ease of visualization, but analyses used trial-level pumping and explosions. In **c** & **d**, points reflect individual participant parameters for loss aversion and beliefs. In all plots, lines represent simple linear regression fits. Grey shading indicates the 95% confidence intervals around the fitted mean. All statistical tests were two-sided and p-values were adjusted for multiple comparisons using the Benjamini-Hochberg procedure.

### Socioeconomic Status Correlated with BELT Performance

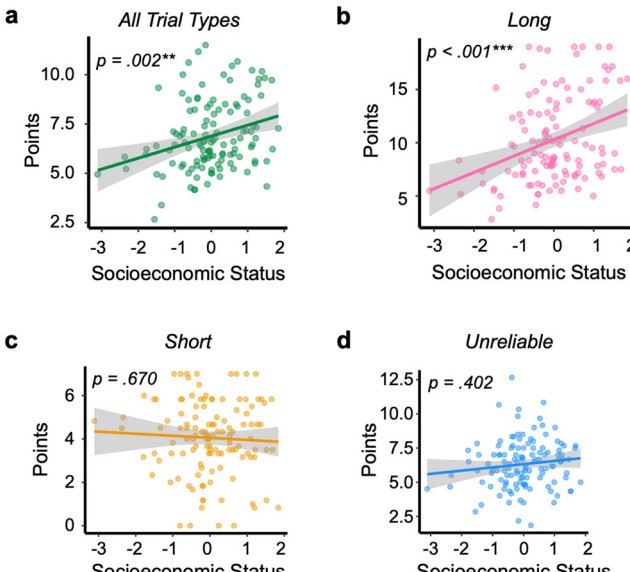

**Fig. 3 | Socioeconomic Status Correlated with BELT Performance (n = 124). a** Higher socioeconomic status (SES) correlated with more points earned in the final third of the task. Statistical test: Mixed-effects negative binomial regression (z = 3.45, p < 0.001, p-adjusted = 0.002, b = 0.084, 95% CI [0.036, 0.132]). **b** This relationship was observed when analyses were restricted to long balloon trials. Statistical Test: Mixed-effects negative binomial regression (z = 4.31, p < 0.001, p-adjusted <0.001, b = 0.159, 95% CI [0.087, 0.232]). **c, d** There was no statistically significant relationship between SES and points earned on unreliable (z = 1.04, p = 0.297, p-adjusted = 0.402, b = 0.039, 95% CI [-0.034, 0.113]) or short balloon trials (z = −0.51, p = 0.612, p-adjusted = 0.670, b = −0.023, 95% CI [−0.110, 0.065]) where exploration was less advantageous. Statistical Tests: Mixed-effects negative binomial regressions. For ease of visualization, in **a–d**, points represent participant-level means (the mean number of points earned). However, all analyses used trial-level data. Lines show simple linear regression fits to participant means. In a-d, grey shading indicates the 95% confidence interval around the fitted mean. All statistical tests were two-sided and p-values were adjusted for multiple comparisons using the Benjamini-Hochberg procedure.

explode when pumped. Using a computational model designed to isolate these factors[57], we found that lower SES was associated with higher loss aversion (t(122) = −2.95, p = 0.004, p-adjusted = 0.007, ρ = −0.26, 95% CI [−0.42, −0.08]; Fig. 2c). There was no statistically significant association between SES and initial beliefs about the probability that the balloon would explode when pumped (t(122) = 0.432, p = 0.667, p-adjusted = 0.697, ρ = 0.04, 95% CI [−0.14, 0.22]; Fig. 2d). Similarly, there was no statistically significant association between SES and belief updating (t(122) = 1.79, p = 0.076, ρ = 0.16, 95% CI [−0.02, 0.33]) risk preference (t(122) = −0.32, p = 0.751, ρ = −0.03, 95% CI [−0.21, 0.15]), or the consistency of decisions (t(122) = 1.34, p = 0.184, ρ = 0.12, 95% CI [−0.06, 0.30]).

#### Loss aversion mediated SES-based differences in exploration
Next, we investigated whether loss aversion mediated SES-based differences in exploration. Across the sample, higher loss aversion correlated with less pumping (t(122) = -8.97, p < 0.001, p-adjusted <0.001, r = −0.63, 95% CI [−0.73, −0.51]) and fewer explosions on unreliable balloon trials (t(122) = −9.98, p < 0.001, p-adjusted <0.001, r = −0.67, 95% CI [-0.76, -0.56]). Loss aversion also fully mediated

SES-based disparities in exploratory pumping (mediation effect [ab] = 0.201, 95% CI [0.097, 0.320], p < 0.001, p-adjusted <0.001) and explosions (mediation effect [ab] = 0.218, 95% CI [0.108, 0.340], p <0.001, p-adjusted < 0.001). Indeed, while lower SES was related to less pumping (total effect [c] = 0.292, 95% CI [0.124, 0.460], p = 0.001) and fewer explosions (total effect [c] = 0.275, 95% CI [0.107, 0.450], p = 0.002), these relationships were not statistically significant after accounting for loss aversion (direct effect on pumping [c'] = 0.091, 95% CI [−0.058, 0.230], p = 0.226; direct effect [c'] on explosions = 0.057, 95% CI [−0.080, 0.190], p = 0.423). Thus, greater loss aversion explained (at least statistically) why lower-SES adolescents explored less.

#### Lower SES Correlated With Reduced BELT Performance
While exploitation might minimize losses on individual trials, this strategy would also be predicted to lead to less feedback about the balloon explosion limits, reducing the ability to learn and optimize performance. To test whether SES was related to performance in the BELT, we focused on the final phase of the task after adolescents had an opportunity to optimize decisions, though, note, results were unchanged when focusing on the full length of the task. We found that lower SES was associated with fewer points earned in the last third of the task (z = 3.45, p < 0.001, p-adjusted = 0.002, b = 0.084, 95% CI [0.036, 0.132]; Fig. 3a). This relationship was also observed on long

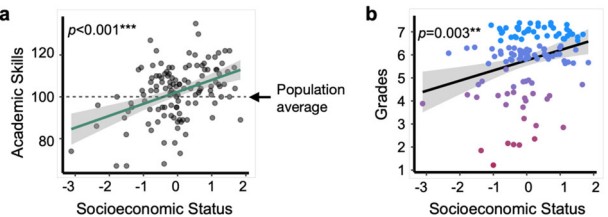

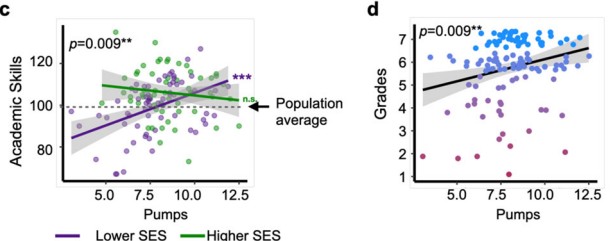

**Fig. 4 | Exploration correlates with socioeconomic disparities in academic achievement. a** Higher socioeconomic status (SES) correlated with higher standardized academic skills (*n* = 124). Statistical Test: Pearson correlation: t(122) = 4.93, *p* < 0.001, *p*-adjusted <0.001, *r* = 0.407, 95% CI [0.249, 0.545]. The dashed line indicates the population average for academic skills. **b** Higher SES correlated with better school grades (*n* = 122). Statistical Test: Cumulative Link Model: (z = 3.22, *p* = 0.001, *p*-adjusted = 0.003, b = 0.61, 95% CI [0.243, 0.990]). **c** Higher exploration, measured by the mean number of pumps on unreliable balloon trials, correlated with higher academic skills (*n* = 124). Statistical test: Linear regression: t(122) = 2.74, *p* = 0.007, *p*-adjusted = 0.009, *r* = 0.241, 95% CI [0.067, 0.399]. This relationship was observed for lower-SES adolescents (purple line; *n* = 66; t(64) = 3.92, *p* < 0.001, *p*-adjusted = 0.001, β = 0.44, 95% CI [0.220, 0.660]) but not higher-SES adolescents (green line; *n* = 58; t(56) = −1.01, *p* = 0.315, *p*-adjusted = 0.402, β = −0.13, 95% CI [−0.400, 0.130]). **d** More pumping correlated with higher grades (*n* = 122). Statistical test: Cumulative Link Model: (z = 2.73, *p* = 0.006, *p*-adjusted = 0.009, b = 0.26, 95% CI [0.076, 0.455]). Each data point represents a participant's academic skill score (**a**, **c**) or grade (**b**, **d**). Lines represent linear regression fits; grey shading indicates 95% confidence intervals around the fitted mean. In **b** & **d**, the color gradient (pink to blue) represents lower to higher grades on a 1–7 scale. All statistical tests were two-sided. *P*-values were adjusted for multiple comparisons using the Benjamini-Hochberg procedure.

balloon trials, where exploration was most beneficial (z = 4.31, *p* < 0.001, p-adjusted <0.001, b = 0.159, 95% CI [0.087, 0.232]; Fig. 3b). In contrast, there was no statistically significant association between SES and points earned on short balloon trials (z = −0.51, *p* = 0.612, p-adjusted = 0.670, b = -0.023, 95% CI [−0.110, 0.065]) or unreliable balloon trials (z = 1.04, *p* = 0.297, p-adjusted = 0.402, b = 0.039, 95% CI [−0.034, 0.113]; Fig. 3c-d). We also observed a relationship between higher SES and better incremental learning: higher SES was linked to steeper gains in points earned across time on long balloon trials (SES x trial number: z = 2.18, *p* = 0.030, *p*-adjusted = 0.045, b = 0.002, 95% CI [0.000, 0.003]). In contrast, we did not observe a statistically significant association between SES and points earned across time on short (SES x trial number: z = −0.58, *p* = 0.564, p-adjusted = 0.648, b = −0.001, 95% CI [−0.005, 0.003]) or unreliable balloon trials (SES x trial number: z = −0.01, *p* = 0.992, p-adjusted = 0.992, b = −0.000, 95% CI [−0.003, 0.003]). Thus, higher SES was linked to better learning and performance in the condition in which exploration was most beneficial.

## Exploration Mediates SES-based Disparities in BELT Performance
We next examined whether exploratory behavior mediated SES-related differences in task performance in the final third of the task.

We focused on points on long balloon trials because these trials drove the task performance differences across SES. We found that exploratory pumping (mediation effect [ab] = 0.154, 95% CI[0.061, 0.260], *p* = 0 .001, *p*-adjusted = 0.003) and explosions on unreliable balloon trials (mediation effect [ab] = 0.121, 95% CI[0.042, 0.220], *p* = 0.001, *p*-adjusted = 0.003) mediated the relationship between SES and points earned. While SES was related to points earned in both models (total effect [c] on points in the pumping model = 0.340, 95% CI[0.172, 0.500], *p* < 0.001; total effect [c] on points in the explosion model = 0.340, 95% CI[0.171, 0.50], *p* < 0.001), this effect was reduced when accounting for pumping (direct effect [c'] = 0.189, 95% CI[0.038, 0.33], *p* = 0.015) and explosions (direct effect [c'] = 0.219, 95% CI[0.063, 0.370], *p* = 0.006). This pattern held when operationalizing performance or exploration using all trial types; Supplementary Note 3). Therefore, reduced exploration partially accounted for why lower-SES adolescents exhibited poorer task performance.

## Exploratory behavior covaries with points earned across time
Given the theoretical and observed importance of exploration for learning, we conducted an exploratory analysis to test whether exploratory tendencies shifted in response to changes in points earned across time. As in prior research[58], we calculated an exponentially weighted moving average of points to capture shifts in reward outcomes across the task. We found that a history of more points earned preceded more pumping (z = 5.46, *p* < 0.001, b = 0.041, 95% CI [0.026, 0.055]) and a higher explosion likelihood (z = 10.36, *p* < 0.001, b = 0.38, 95% CI [0.33, 0.50]). SES was not a statistically significant moderator of these relationships (SES x pumping: z = −1.69, p = 0.090, b = −0.012, 95% CI [−0.026, 0.009]; SES x explosions: z = 0.08, *p* = 0.932, b = 0.003, 95% CI [−0.090, 0.070]). Thus, more points preceded more exploratory behavior, possibly mirroring the impact of real-world resources on exploratory decisions.

## Exploration relates to socioeconomic disparities in academic achievement
As in previous studies[2,4], lower SES correlated with reduced academic skill scores (t(122) = 4.93, *p* < 0.001, *p*-adjusted <0.001, *r* = 0.407, 95% CI [0.249, 0.545]; Fig. 4a) and lower grades (z = 3.22, *p* = 0.001, *p*-adjusted = 0.003, b = 0.61, 95% CI [0.243, 0.990]; Fig. 4b). Additionally, more pumping on unreliable balloon trials correlated with better academic skills (t(122) = 2.74, *p* = 0.007, *p*-adjusted = 0.009, *r* = 0.241, 95% CI [0.067, 0.399]; Fig. 4c) and higher grades (z = 2.73, *p* = 0.006, *p*-adjusted = 0.009, b = 0.26, 95% CI [0.076, 0.455]; Fig. 4d). By contrast, there was no statistically significant correlation between the mean number of explosions and academic skills (t(122) = 1.46, *p* = 0.146, *p*-adjusted = 0.146, *r* = 0.131, 95% CI [−0.046, 0.301]) or grades (z = 1.619, *p* = 0.105, *p*-adjusted = 0.120, b = 2.071, 95% CI [−0.414, 4.610]). Thus, greater exploration correlated with better academic achievement for one measure of exploration but not the other.

The positive association between exploratory pumping and academic skills was strongest among lower-SES adolescents (SES x pumping: t(120) = −2.79, *p* = 0.006, *p*-adjusted = 0.010, β = −0.20, 95% CI [−0.340, −0.060]; Fig. 4c). In contrast, SES did not statistically significantly moderate the relationship between pumping and grades (z = −0.602, *p* = 0.547, *p*-adjusted = 0.648, β = −0.08, 95% CI [−0.360, 0.190]. To further probe how pumping related to academic skills across SES, we divided the sample into a lower- and higher-SES sub-sample using a mean split on the SES data. The higher-SES group comprised 58 participants (median income: $150,000; range: $65,000–$1.25 million; median parental education: 17 years; range: 14–20 years). The lower-SES group comprised 66 participants (median income: $70,000; range: $2,000–$200,000; median parental education: 13 years; range: 7–17 years). We found that more pumping correlated with better academic skills in the lower-SES sub-sample (t(64) = 3.92, *p* < 0.001, *p*-adjusted = 0.001, β = 0.44, 95% CI [0.220,

0.660]; Fig. 4c). In contrast, there was no statistically significant association between pumping and academic skills in the higher-SES sub-sample (t(56) = −1.01, *p* = 0.315, *p*-adjusted = 0.402, β = −0.13, 95% CI [−0.400, 0.130]; Fig. 4c).

Next, we tested whether exploratory pumping mediated SES-based disparities in grades and academic skills. For academic skills, we focused on the lower-SES subsample since there was no statistically significant association between exploratory pumping and academic skills in the higher-SES subsample. In lower-SES adolescents, exploratory pumping fully mediated SES-based differences in academic skills (mediation effect [ab] = 0.194, 95% CI[0.047, 0.39], *p* = 0.002, *p*-adjusted = 0.005). While higher SES was related to better academic skills (total effect [c] = 0.466, 95% CI[0.133, 0.790], *p* = 0.008), this relationship was not statistically significant after accounting for pumping (direct effect [c′] = 0.272, 95% CI[−0.066, 0.600], *p* = 0.108). Furthermore, across the entire sample, pumping partially mediated SES-based differences in grades (mediation effect [ab] = 0.054, 95% CI[0.004, 0.130], *p* = 0.035, *p*-adjusted = 0.050). While SES positively correlated with grades (total effect [c] = 0.291, 95% CI[0.113, 0.460], *p* = 0.001), this relationship was significantly reduced when accounting for pumping (direct effect [c′] = 0.237, 95% CI[0.052, 0.410], *p* = 0.013). These relationships were maintained when using different operationalizations of exploration (Supplementary Note 4). Note that there was no statistically significant association between loss aversion and SES-based disparities in academic outcomes (Supplementary Note 5). Taken together, these findings link exploratory behavior to socioeconomic disparities in academic achievement.

## Discussion

The present study examined the relationship between SES, task-based exploration, and real-world academic achievement in 12–14 year-old adolescents. We observed that lower SES correlated with less exploration in the BELT. Reduced exploration partly mediated the relationships between lower SES and poorer task performance. Furthermore, reduced exploratory pumping mediated the relationship between lower SES and lower grades and academic skills. While this study was purely correlative, which limits causal inference, the patterns align with theoretical predictions that SES shapes exploratory decisions, which, in turn, limit learning and achievement. We also observed that lower SES was linked to higher loss aversion, which mediated SES-based differences in exploration. Notably, however, exploratory behavior was not stable, but fluctuated across the task in response to shifting reward outcomes: more points preceded more exploratory behavior in time, regardless of SES, suggesting environments with abundant rewards might boost exploration. Altogether, these patterns align with theories that individuals adapt decision strategies to match the constraints of their early environment, leading to reduced exploration among lower-SES adolescents. These exploratory decisions correlate with learning and academic achievement, though future studies will be essential to establish causal relationships.

Although lower-SES adolescents exhibited greater exploitation, there was no statistically significant evidence that this strategy conferred advantages in the short balloon condition where exploitation was optimal. This result aligns with findings that early adversity heightens exploitation without necessarily leading to performance advantages in contexts that reward it[43]. A key factor in understanding this pattern may be the distinction between overgeneralizing an exploitative strategy across contexts and flexibly adapting exploratory strategies based on environmental context. Higher-SES adolescents may have learned the optimal strategy for each balloon condition and adapted their strategies accordingly–exploring and exploiting when adaptive to do so. This flexibility would maximize performance in each condition. Conversely, lower-SES adolescents may have generalized an exploitative approach–possibly driven by heightened loss aversion–rather than adapt to the specific demands of each balloon type.

Addressing SES-based disparities in performance may, therefore, require supporting context-appropriate decision strategies that boost exploration when optimal.

The present study not only established a link between lower SES and reduced exploration but also identified loss aversion as a possible underlying mechanism. Lower-SES adolescents may, therefore, feel less equipped to handle the potential costs of exploring. There was no statistically significant evidence that lower-SES adolescents started the task believing that the balloon would explode earlier compared to higher-SES adolescents. This finding contrasts work linking lower SES to greater pessimism[53,54], which presumably could heighten expectations of earlier balloon explosions. Notably, though, the parameter we measured here may not map directly onto these broader assessments of optimism and pessimism. We also found no statistically significant evidence linking lower SES to learning rate, risk preference, or decision consistency. However, other cognitive processes not modeled here could have contributed. For instance, lower SES correlates with steeper temporal discounting[14,16,17,59,60]–a preference for earlier rewards–which might have led to greater exploitation. Future work incorporating temporal discounting into a formal computational model could test this mechanism.

Although reduced exploration may be adaptive in resource-scarce or unpredictable environments[11–14,16,17,43], this strategy may become maladaptive if generalized to safe contexts. Indeed, we found that lower-SES adolescents exhibited greater exploitation, even within a low-stakes laboratory task. Furthermore, one measure of in-task exploration (pumping) correlated with SES-based differences in grades and academic skills. These findings inform emerging perspectives that decision strategies may serve as an underappreciated mechanism driving achievement gaps[11].

Based on these findings, an open question is how to foster adaptive exploratory behavior in lower-SES youth. Our findings point to one possibility: increasing environmental rewards and resources. We observed that fluctuations in rewards in the BELT covaried with exploration on a trial-by-trial basis: Greater rewards preceded greater exploration in time. This dynamic co-fluctuation aligns with research demonstrating that periods of resource abundance boost exploration across species, including human adults[61,62] and adolescents[58], rhesus macaques[63], and even honey bees[64], and can even impact cognitive abilities[65]. Furthermore, negative experiences, like natural disasters causally increase risk aversion[66]. Thus, while individuals may differ in baseline exploratory tendencies due to early environmental tuning, rewards might shift this set point, boosting exploration when rewards are abundant. Supportive environments with ample resources and low stakes for failure might therefore promote positive risk-taking and exploration in learning environments like classrooms.

The present study has several important limitations that could be addressed in future research. First, the study was not pre-registered, a step that enhances transparency and reproducibility in scientific research[67]. Second, while the BELT demonstrates strong test-retest reliability and moderate correlations with personality traits linked to risk-taking[55], its relationship to real-world exploration remains untested. However, the Balloon Analogue Risk Task (BART), from which the BELT was adapted, shows moderate correlations with self-reported risk-taking behaviors in adolescents[68] and adults[69,70], though the strength of these associations varies across measures of risk-taking[70]. Given its derivation from the BART, the BELT may similarly predict real-world exploratory and risk-taking behaviors. However, this hypothesis requires direct empirical validation. Third, null effects reported in this study show absence of evidence rather than evidence of absence, as our statistical tests cannot provide support for null hypotheses without pre-specified equivalence testing. Finally, the cross-sectional design and mediation analyses preclude the establishment of causal inferences[71]. It is our hope that these correlative findings inspire future studies designed to test the causal impacts of SES.

In conclusion, these findings offer insights to educators, policy-makers, and researchers interested in redressing disparities in learning and achievement. By linking SES, exploration, and academic achievement, these findings contribute to theories about why lower-SES adolescents might struggle more to achieve academically. Our findings raise the possibility that disparities in academic achievement may reflect not only the distal limitations of lower-SES environments but also a specific proximal psychological mechanism in the minds of adolescents that may limit learning measured in multiple ways. This research, therefore, calls for a new emphasis on how exploratory decisions might drive learning disparities across SES.

## Methods

### Ethics statement
Participants provided informed assent, and their legal guardians provided informed consent. The study was approved by the MIT Committee on the Use of Human Subjects.

### Participants
One hundred and twenty-nine children and adolescents were recruited through flyers, social media, and local schools in and around the Boston area between 2017 and 2020. Eligible participants were those in the 7th or 8th grade who did not have a reported history of autism spectrum disorder or a neurological disability, were proficient in English, had no MRI contraindications, were not born premature (< 34 weeks) and had an accompanying parent with English and/or Spanish proficiency. Of the 129 participants recruited, 125 completed the BELT. However, one of the 125 participants did not report parental education or income data and was therefore excluded from analyses. The final sample included 124 participants (mean age: 13.46; range: 12.04–14.8 years, 62 female, 62 male). Demographic information, including ethnicity, reading, and language difficulties and delays, can be found in Supplementary Tables 1 and 2.

All participants were renumerated at a fixed rate, regardless of their task performance. This ensured that any potential differences in motivation for monetary rewards would not contribute to SES-based differences in task performance. Participants received $75 for completing the BELT and other cognitive tasks, as well as demographic questionnaires and surveys, and could earn up to $200 for completing all aspects of the study, including MRI tasks not reported here.

We aimed to collect data from at least 100 participants based on moderate to large effect sizes of SES and cognitive performance[1–3]. An a priori power analysis indicated that 100 participants enabled 80% power to identify medium-sized effects (d of 0.57 or Pearson's r of 0.28) in two-tailed individual difference analyses. Furthermore, a sensitivity analysis revealed that 124 participants allowed for 80% power to identify even smaller effects–d of 0.51 or r of 0.248–in two-tailed individual differences analyses.

### Procedure
The BELT was administered as part of a larger study examining the relationship between SES, brain development, cognition, and academic achievement conducted at the Massachusetts Institute of Technology (MIT). All tasks were completed in a quiet testing room. Participants completed the BELT on a laptop computer and the Woodcock-Johnson Test of Achievement (Edition IV) was administered by trained research staff. Parents or caregivers self-reported demographic information, including household income and parental education, the participant's sex, ethnicity, school grades, and whether participants had been diagnosed with a learning disorder. Sex was not considered in the study design and no sex-based analyses were conducted as the study was not powered to detect sex differences and this was not a focus of the research. Note that investigators were blind to participants' SES. Additional study components not reported here

included MRI tasks and other behavioral measures completed during a separate visit to MIT.

### Socioeconomic status
To assess SES, we collected two primary measures from parents. Annual household income (mean: $134,795.9; median: $100,000; range: $2000 to $1.25 million; Fig. 1a) was obtained through a free-text entry and was log-transformed to correct for left skew in the data and to account for the more significant impact of income gains on lower-SES individuals. Parental education levels were collected using a 9-point scale, which we converted to years of formal schooling (Fig. 1b). For participants with two parents/caregivers, we calculated the mean years of schooling across both parents (mean: 15.07 years; median: 15 years; range: 7–20). To create a standardized composite SES metric, we computed z-scores for the log-transformed income and mean parental education levels. The SES composite score was derived by calculating the mean of these two z scores, giving equal weight to household income and parental education measures. For participants with missing data (3 were missing income, 5 were missing education levels for the father/caregiver 2), we calculated SES using the mean of the z scores of the two available measures. Supplementary Fig. 1 shows the SES composite score of the sample.

### Balloon emotional learning task
In this computerized task[43,55,56], participants were instructed to press the spacebar key to pump up a virtual balloon on each trial to earn points. Each time participants pumped the balloon, they earned 1 point, and the balloon grew visually larger on the screen. However, participants could not simply continue to pump the balloon to earn unlimited points; over-inflating the balloon led to a balloon explosion and the loss of all points for a trial. Participants could, therefore, stop pumping and 'save' their accumulated points for a trial at any point by pressing 'enter'. Thus, throughout the task, participants had to balance the desire to secure their accumulated points by ceasing to pump (exploit) against the desire to earn more points by continuing to pump (exploring). Critically, the more participants pumped, the more they learned about the balloons' explosion limits, but the more they risked an explosion.

There were three different types of balloons, each with a unique explosion threshold. These balloons were differentiated by a unique color (blue, pink, orange; Fig. 1b). Before the task, participants were told that some balloons popped after many pumps, whereas some popped after just a few. However, participants were not told the popping threshold of each balloon. Instead, participants had to learn these thresholds through experience. There was an unreliable blue balloon that exploded after a variable number of pumps (8, 14, or 20 pumps), a long pink balloon that exploded at 20 pumps (favoring a more exploratory strategy), and a short orange balloon that exploded at 8 pumps (favoring a more exploitative strategy). Across 54 trials, participants could learn, through pumping and experiencing explosions, the optimal pumping strategy for each balloon. Each balloon was presented six times across each third of the task.

As in previous research[43,55], we measured exploration using the mean number of pumps and explosions. We selected pumping as an index of exploration because the more participants pumped, the more they were willing to embrace uncertainty and gather information about the balloons' explosion limits. Furthermore, explosions reflected the consequences of exploring beyond the explosion thresholds, thereby serving as a proxy for exploration and critical feedback about the limits of each balloon condition. Task performance was measured as the mean number of points earned.

### Grades
Parents reported participants' grades on a categorical scale, selecting from mostly As, a mix of As and Bs, mostly Bs, a mix of Bs and Cs,

mostly Cs, a mix of Cs and Ds, and mostly Ds or below. We converted grades to a numerical scale of 1–7 (1 being the lowest, 7 being the highest). Data were missing for 2 participants, leaving 122 participants in analyses involving grades.

## Academic skills

Participants completed the Academic Skills composite of the Woodcock-Johnson Test of Achievement (Edition IV). This composite combines subtests of word reading, spelling, and mathematical calculation skills. Scores are evaluated against a normative sample stratified by age and grade.

## Statistical analyses

Statistical analyses were performed in R (version 4.2.2). Prior to fitting models, we assessed relevant model assumptions through visual inspection of residuals (Q-Q plots, residuals versus fitted values) and formal tests (e.g., Shapiro-Wilk test for normality in correlations, simulation-based residuals for count data). When assumptions of normality or equal variances were violated, we used alternative models.

For correlation analyses, we fit Pearson correlations if Shapiro-Wilk tests confirmed normality and Spearman rank-order correlations otherwise. Count outcomes were modeled with negative binomial regressions using the glmmTMB package to address overdispersion. Ordinal data were modeled with cumulative link models (logit link) and binomial outcomes were modeled using logistic regressions (lme4 package). For repeated measures predictors, we fit mixed-effects models with random intercepts per participant and random slopes for repeated within-participant variables. Models that did not converge were simplified iteratively (e.g., removing random slopes) until they converged[72]. Prior to fitting models, binary predictors were effect-coded and continuous predictors were mean-centered.

All variables included in mediation analyses were z-scored prior to model fitting to obtain standardized effects. Mediation models involved 5000 bootstrap iterations to estimate direct, indirect, and total effects. For the ordinal grades variable (7-point scale), we treated its z-scored version as continuous in linear models after confirming via residual diagnostics that assumptions of homoscedastic variance and approximate normality were met. This approach is supported by methodological guidelines demonstrating minimal bias when ordinal variables have at least five categories and exhibit a roughly symmetric distribution[73].

To control for multiple comparisons, we implemented false discovery rate (FDR) corrections separately for three conceptually distinct families of analyses reported in the main text. The first family focused on learning on the BELT, which included analyses relating trial number to points earned. The second family of tests focused on how exploration was related to other variables including task performance (points) and loss aversion. The third family of tests included analyses relating SES to other variables, including exploratory behavior, BELT performance, and academic outcomes. For each family, p-values were ranked from smallest to largest and compared against critical values calculated using the Benjamin-Hochberg procedure. We report both uncorrected and FDR-adjusted p-values in the text. Note that exploratory and validation analyses were not corrected for multiple comparisons. All statistical tests of significance were two-tailed. P-values ≤ 0.05 were considered statistically significant. All models included 124 participants except for analyses involving grades, which included 122 participants.

## Examining incremental learning across the BELT

As in prior research[43,55], we anticipated that adolescents would earn more points as they gained experience with the task, reflecting incremental learning. To test this, we fit a mixed-effects negative binomial regression (appropriate for count data) in which trial-by-trial points served as the DV and trial number served as the IV (i.e., points ~ trial number). We fit an analogous model in which balloon type was included as an IV and interaction term with trial number (i.e., points ~ trial number x balloon type) to examine whether participants displayed stronger incremental learning on long and short than unreliable balloon trials. In the supplement, we report on how trial number was related to the number of pumps and explosions (Supplementary Note 1). In these mixed-effects negative binomial regressions, condition-level effects were extracted by re-levelling the base variable.

## Probing whether exploration boosted performance on the BELT

We reasoned that if exploration boosted learning, greater exploration early in the task should lead to better performance later. To test this idea, we fit two Partial Spearman correlations examining whether a greater mean number of pumps or explosions in the first third of the task predicted a greater mean number of points earned in the last third of the task. Spearman correlations were used to account for non-normal distributions of model residuals. We controlled for the mean number of points earned in the first third of the task to ensure relationships were unrelated to early knowledge of the balloon-explosion contingencies (i.e., we performed a Granger Causality Analysis). The mean number of pumps and explosions served as measures of exploration in separate models.

## Probing the relationship between SES and exploration

To assess individual differences in exploration, we used the number of pumps and explosions on unreliable balloon trials. We selected unreliable balloon trials a priori because we anticipated that behavior on the other trial types would be confounded by learning effects (See Results and Supplementary Note 1). Split-half reliability analyses confirmed that both metrics were stable: the mean number of pumps on odd and even trials were strongly correlated across participants ($t(122) = 20.09$, $p < 0.001$, $r = 0.88$, 95% CI [0.833, 0.914]), as were the mean number of explosions ($t(122) = 6.20$, $p < 0.001$, $r = 0.49$, 95% CI [0.343, 0.613]). Thus, individuals who were highly exploratory at some time points were similarly exploratory at others.

One negative binomial regression and one logistic regression model were fit to examine the relationship between SES and pumping and explosions. SES served as the IV, and the number of pumps and explosions on unreliable balloon trials served as DVs in separate models (i.e., pumps ~ SES; explosions ~ SES). For thoroughness, we repeated these analyses, operationalizing exploration as the number of pumps and explosions across all trials (Supplementary Note 2), which did not change the pattern of results.

## Computational model of decision making

We fit a computational model to the pumping data using the Exponential-Weight Mean-Variance model from the 'hBayesDM' package in R[57]. This model extracts parameters reflecting psychological processes underlying choices to pump or save points. This model includes five parameters, but we were particularly interested in parameters representing loss aversion and initial beliefs about the balloon bursting. The loss aversion parameter represents how much more psychologically impactful a potential loss is compared to an equivalent gain on each individual decision to pump the balloon. Higher values indicate how much more heavily losses are weighed relative to equivalent gains when making the decision to pump or not. The parameter representing prior beliefs of a burst represents a participant's initial subjective probability that pumping will make the balloon burst, before any task experience. Higher values indicate that participants start the task by assigning a higher subjective probability of the balloon bursting when pumped. This initial belief parameter gets dynamically updated through experience during the task via the updating exponent parameter, which controls how quickly participants modify their beliefs about the balloon bursting based on

observed outcomes. For the updating exponent, higher values represent more rapid adjustments in response to recent outcomes. The model also includes two other parameters: risk preference, representing an individual's general propensity towards or aversion to risk, where positive values indicate risk aversion; inverse temperature, reflecting the consistency or randomness of decision-making, where higher values reflect more deterministic decisions.

The model calculates the subjective utility for each pump, balancing the expected reward against the risk of bursting, captured by the following equation:

$$U_{kl}^{pump} = (1 - p_k^{burst})r - p_k^{burst}\lambda(l-1)r$$
$$+ \rho p_k^{burst}(1 - p_k^{burst})\{r + \lambda(l-1)r\}^2 \text{ with } \lambda > 0, U_{kl}^{transfer} = 0,$$
(1)

Where $U_{kl}^{pump}$ denotes the subjective utility for each pump, $l$, on trial $k$; $p_k^{burst}$ is the perceived probability of the balloon bursting if pumped, $r$ is the amount of reward for each successful pump, $\lambda$ is a parameter representing loss aversion, the degree to which a balloon bursting reduces pumping, $\rho$ is the risk preference indicating an individual's propensity towards exploration, and $l$ indexes the number of pumps during a trial. The formula also includes a term for the utility of transferring (not pumping further), set to zero $U_{kl}^{transfer} = 0$, emphasizing the decision-making process between continuing to pump to increase potential rewards versus transferring the accumulated reward to avoid the risk of loss. The model employs Markov Chain Monte Carlo (MCMC) sampling with 4000 iterations across 4 chains, discarding the first 1000 iterations as warm-up to ensure parameter estimate reliability.

**Evaluating the computational model fit to the data.** All parameters demonstrated excellent convergence (Rhat of 1.0) and sufficient effective sample sizes (all ESS > 1750). We conducted posterior predictive checks by simulating data from the fitted model and compared them to observed pumping behavior. The model accurately replicated participant behavior, showing a strong correlation between predicted and actual pumps (Pearson's correlation: t = 431.12 (6694), $p < 0.001$, r = 0.98, Supplementary Fig. 2).

**Correlating socioeconomic status with model parameters**
After extracting model parameters, we fit 5 Spearman correlation models to account for the non-normal distribution of model residuals. In these models, SES served as the IV, and each of the 5 model parameters served as the DV in separate models. Models focusing on loss aversion and initial beliefs about how early balloons would pop were confirmatory tests. We also fit two bootstrap mediation models to test whether loss aversion mediated SES-based differences in exploratory pumping and explosions.

**Probing the relationship between SES and performance in the BELT**
We fit a mixed-effects negative binomial regression in which SES served as the IV and trial-level points served as the DV (i.e., points ~ SES). We focused on points in the last third of the task to capture final performance outcomes, though results were unchanged when focusing on points across the whole task. This model included all trial types, but we fit 3 additional, identical models, restricted to the long, short, and unreliable balloon trials. We also tested whether the degree to which performance improved across the task (in terms of points earned) differed by SES. In these models, trial-level points served as the DV and SES and trial number served as IVs and interaction terms.

We fit two bootstrap mediation models to test whether exploration contributed to SES-related disparities in task performance. The mean number of pumps and explosions on unreliable balloon trials served as mediators for the relationship between SES and final performance outcomes (points in the last third of the task). We focused on the long balloon trials, as only these trials exhibited SES-related differences in performance (See Results).

**Do rewards boost exploration?**
To understand whether individuals flexibly adapted exploration based on recent outcomes, we explored whether fluctuations in points earned across time covaried with exploration on a trial-by-trial basis. For each trial, we calculated an exponentially weighted moving average (EWMA) of points, reflecting an individual's recent history of points earned using the following update rule:

$$EMWA_t = \alpha \times r_t + (1 - \alpha) \times EMWA_{t-1}$$
(2)

Here, $EWMA_t$ represents the EWMA at the current trial, $t$, $\alpha$ is the smoothing factor or learning rate parameter that determines the influence of the most recent observation on the moving average, and $r$ represents the reward on the current trial, $t$. The learning rate parameter, $\alpha$, was calculated as $2/(N+1)$, where $N$ was set to 10.

We then fit a logistic regression and a negative binomial regression model, in which the trial-by-trial EWMA served as the IV, and either trial-by-trial pumps or explosions on subsequent trials served as the DV. We included SES as an IV and interaction term to test whether the relationship between reward and exploration differed by SES (i.e., pumps ~ moving average of reward x SES; explosions ~ moving average of reward x SES). Trial number was included as a covariate to ensure any relationships were unrelated to the EWMA and exploration independently increasing across the task. To ensure relationships did not reflect immediate decreases in exploration after an explosion, we included previous trial explosions as a covariate.

**SES, exploration, and academic achievement**
We fit a cumulative link model and a linear regression model to test whether SES was related to grades and academic skills. SES served as an IV in both models, and grades and academic skills served as DVs in separate models (i.e., grades ~ SES; academic skills ~ SES). We then tested whether exploration correlated with academic achievement using linear regression models. The mean number of pumps and explosions on unreliable balloon trials served as IVs in separate models, and grades and academic skills as DVs in separate models. We fit additional models that included SES as a covariate and interaction term. Significant interactions were probed by re-fitting models separately for a higher- and lower-SES subgroup, divided using a mean split on the SES data ($n = 58$ in the higher SES group [median income: $150,000; range: $65,000–$1.25 million; median parental education: 17 years; range: 14–20 years, $n = 28$ male, $n = 30$ female] and 66 in the lower SES group [median income: $70,000; range: $2000-$200,000; median parental education: 13 years; range: 7–17 years, $n = 34$ male, $n = 32$ female]).

Finally, we fit 2 bootstrap mediation models to test whether pumping on unreliable trials mediated the relationship between SES and grades or academic skills. Two exploratory mediation models were fit to assess whether loss aversion mediated the relationship between lower SES and reduced grades and academic skills (Supplementary Note 5).

**Reporting summary**
Further information on research design is available in the Nature Portfolio Reporting Summary linked to this article.

## Data availability
Raw and processed data for this article is publicly available on the Open Science Framework at the following link[74]: https://osf.io/hwt6z/.

## Code availability
The code for this article can be found on the Open Science Framework at the following link[74]: https://osf.io/hwt6z/.

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

## Acknowledgements

This research was supported by a Natural Sciences and Engineering Research Council of Canada Postdoctoral Fellowship and a Banting Postdoctoral Fellowship (A.L.D.) and the William and Flora Hewlett Foundation [#4429 (J.D.E.G.)] N.A.H. was partially supported by the Brain and Behavior Research Foundation (#27970) and the NIGMS (P20GM130461-6206).

## Author contributions

J.L., R.R., N.A.H., C.C.C.B., and J.D.E.G. designed the study; R.R., H.G., M.A.G., Y.C.T., A.I. collected the data; A.L.D. conceptualized and performed the analyses, made the figures and wrote the first draft of the paper. J.L., R.R., J.I., N.A.H. and J.D.E.G. provided critical revisions. All authors approved of the final manuscript. J.D.E.G. provided resources and supervision.

## Competing interests

The authors declare no competing interests

## Additional information

**Peer review information** *Nature Communications* thanks Willem Frankenhuis and the other, anonymous, reviewer for their contribution to the peer reviewer(s) of this work. A peer review file is available.

