## [Transparent Peer Review file · Nature Communications]

Exploration is Associated with Socioeconomic Disparities in Learning and Academic Achievement in Adolescence

Corresponding Author: Dr Alexandra Decker

Version 0:

Reviewer comments:

Reviewer #1

(Remarks to the Author)

Thank you for the opportunity to review this manuscript. In this study, the authors examine the association between explore/exploit preferences, socioeconomic status (SES) and academic attainment in a sample of 12-14 year olds. They find participants from lower SES backgrounds explore less than peers from higher SES backgrounds; further, lower rates of exploration were associated with lower academic attainment. This is a timely topic bridging the explore/exploit framework to real world decision-making, which thus far has been lacking in the literature. However, I feel this study has some important caveats that are not sufficiently acknowledged in the present manuscript and would need to be addressed before it is suitable for publication. Most pressingly, I think there needs to be greater acknowledgement in the text that this is cross sectional data which is an important caveat given the authors propose these findings might be used to guide policy. I have some additional comments/queries, which I outline by section below:

Introduction:

For a more general readership, it might be helpful to expand on the summary of why exploration in childhood and adolescence is beneficial later in life. The authors cite relevant work from Gopnik and colleagues – summarizing their account in slightly greater detail would provide greater context for the importance of considering why reduced exploration may have downstream consequences in the lifespan.

In discussing the survival benefits of adjusting explore/exploit strategies to the availability of rewards in the environment, I suggest referring to non-human animal research that has provided stronger causal evidence for this account (see Pradhan et al. 2019; also reviewed in Lloyd et al., 2023).

The authors do not state any explicit hypotheses for the present study, though they review sufficient literature to propose directional predictions about the association between SES, exploration and academic attainment. I am not suggesting the authors should hypothesise after results are known, but it would be helpful to understand whether the authors did have specific predictions in order to understand whether the data are able to test these predictions.

Methods:

There is important demographic information missing about the participants recruited for this study, including their ethnicity, English language ability, and any developmental disorders that could impact language skills (the latter of these points may be covered under the neurological conditions assessed, but it would be helpful to clarify). This consideration is important for several reasons:

- i) Such information would be necessary to report to ensure the study is reproducible.
- ii) Given evidenced SES differences responses to opt-in parental consent (Spence et al., 2015), it would be important to report demographic information to consider whether the sample was representative of low- and high-SES populations.
- iii) The measures of academic attainment are highly reliant on English language ability. It would be important to understand whether any students had diagnoses that could impair their written or spoken language, which may impact measures of academic attainment.

Could the authors clarify whether participants were remunerated at a single rate, or whether payment was performance-dependent? It would be helpful for the authors to provide a brief rationale for the strategy they adopted, as performance-dependent payment may increase motivation, but may also covary with SES.

Results:

Greater a priori justification is needed for the choice of predictors and outcome variables in the regression and mediation analyses. The authors use pumps in the unreliable balloon trials in their analyses to test the hypothesis related to SES and exploration, but then restrict their analyses to the long pump condition when testing whether exploration mediates the association between SES and task performance. In the absence of a pre-registered analysis plan, it is difficult to understand which of these analyses were planned prior to examining the data.

Given that multiple analyses are conducted on the same dataset, I suggest correcting the analyses for multiple comparisons.

The authors should be commended for the thorough analyses of the task data. However, not all of the findings presented in the results section appear to be directly related to their research question. For example, the analysis of improved learning across the task is helpful to establish the validity of the task in this age group but does not directly relate to the main research question.

If loss aversion is the proposed mechanism driving reduced exploration (and subsequent poorer academic attainment) in individuals from low SES backgrounds, should this parameter not also be subject to similar analyses to exploration? I.e., analysis to examine whether loss aversion mediates the association between SES and academic attainment. Such analysis should be labelled as exploratory but could provide more precise insight into how interventions may address SES disparities in education, should these findings be replicated in a longitudinal sample.

It would be helpful to provide information about the number of participants assigned to the low SES and high SES categories after the mean split was conducted.

Discussion:

It would be worth noting that the authors did not find evidence for improved performance in the short pump threshold condition for adolescents from low SES backgrounds. This finding is in contrast to Humphreys et al. (2016) who found early adversity conferred benefits in low threshold conditions. While the operationalization of early life experience (adversity versus socioeconomic status) is different between the studies, it would be helpful to frame the findings in context of this previous research if the authors are to consider the implications for educational settings.

The authors should discuss limitations to the present study. Most importantly, it would be necessary to highlight that the cross-sectional data precludes any causal inference being drawn from this study and I think a greater recognition of this limitation throughout the discussion (e.g., in the summary of the results in the first paragraph of the discussion) would provide a more accurate summary of the contribution of this study to the literature. Discussing the limitations of the design would also be important when reflecting on the choice of analyses, as mediation models are not always appropriate for cross-sectional data (see Cain et al., 2018). At the very least, this limitation should be discussed with relation to the conclusions that can be drawn from these data and the type of analysis used.

Minor:

p.3 line 74 missing the word 'theories' after 'prominent'

References:

- Cain, M. K., Zhang, Z., & Bergeman, C. S. (2018). Time and other considerations in mediation design. *Educational and Psychological Measurement*, 78(6), 952-972.
- Lloyd, A., Viding, E., McKay, R., & Furl, N. (2023). Understanding patch foraging strategies across development. *Trends in Cognitive Sciences*.
- Pradhan, S., Quilez, S., Homer, K., & Hendricks, M. (2019). Environmental programming of adult foraging behavior in *C. elegans*. *Current Biology*, 29(17), 2867-2879.
- Spence, S., White, M., Adamson, A. J., & Matthews, J. N. (2015). Does the use of passive or active consent affect consent or completion rates, or dietary data quality? Repeat cross-sectional survey among school children aged 11–12 years. *BMJ open*, 5(1), e006457.

(Remarks on code availability)

Given my reservations about the study in its current form, I have opted not to review the code on this occasion and hope this is okay. If the authors are invited to revise and resubmit the paper, I would be very happy to review the code to ensure it is computationally reproducible.

Reviewer #2

(Remarks to the Author)

This well-written paper is theoretically interesting and timely, and will likely be of interest to the readership of *Nat Comm*. It certainly has the potential to make a valuable contribution. However, the paper could be improved along several dimensions. Below I list 8 major points, all of which can be resolved, and several minor points. I would look forward to reading a revised version of this paper.

MAJOR POINTS

1. Data and script

I appreciate the authors sharing their data and script. However, in the data file, I was unable to find descriptions of the variables (only labels); and the script included minimal annotation. To ensure reproducibility, it would be good to improve these aspects of documentation.

2. Preregistration

The paper reports confirmatory research (line 623); that is, it uses data to test hypotheses, rather than to generate hypotheses (as done in exploratory research). However, I could not find a mention of preregistration (<https://doi.org/10.1073/pnas.1708274114>). If the research was not preregistered, it would be good to make this explicit (a) when introducing the current study (e.g., on line 135, the authors could add: "The current research was not preregistered.") and (b) to mention it as a limitation in the Discussion section.

3. Methods and participants

The information on methods and participants seems to be incomplete (and consistent in one place). Please specify the place(s) from which the sample was drawn (e.g., country, region, or city). It would be good to include this information in the abstract and in the "Participants" section in the main text. Please specify the test setting (e.g., online, lab, community center). Third, the number of participants included in the analyses of grades is initially described as 114 (line 393) and later as 115 (lines 521-522). Fourth, the upper end of the range of household income is initially described as "1.1 million" (line 137) and later as "\$1.25 million" (line 476). Fifth, it would be valuable to provide more information about the distributions of both the component and the composite SES measures (in a table or figure in the Supplementary Materials).

4. Calibrated conclusions

There is scope to better calibrate conclusions to the data. The authors compared two measures of exploration (number of pumps and explosions) with two measures of academic achievement (grades and academic skills). The data support associations between pumping and both measures of academic achievement, but do not support associations between explosions and either measure of academic achievement. This is described on lines 356-362: "more pumping on the BELT correlated with better academic skills (...), although the mean number of explosions were unrelated to academic skills (...). More pumping (...), although not more explosions (...) on unreliable balloon trials predicted higher grades." Despite the evidence being mixed, the authors proceed as though the evidence provides consistent support. For instance, the quoted paragraph concludes with: "Thus, greater exploratory tendencies correlated with better academic achievement." In my view, this conclusion would be better calibrated if it read: "Thus, greater exploratory tendencies correlated with better academic achievement on some measures, but not others". Similar statements occur later in the paper, for instance, on lines 445-445: "we observed that reduced exploratory tendencies were related to lower academic achievement". The paper could do more justice to the mixed nature of the evidence.

5. Interpretation of null results

196-198: "we used unreliable balloon trials to quantify individual differences in exploration, as these trials were immune to learning related adjustments behavior" — strictly speaking, the analyses support the conclusion that there is no evidence for learning (absence of evidence), but not the conclusion that there is evidence of no learning (evidence of absence). To establish the latter, one would need to conduct analyses that can test support "for" the null hypothesis (rather than only "against" it), such as equivalence testing (<https://doi.org/10.1177/2515245918770963>). The same issue applies to the discussions of exploration not boosting performance on the short balloon trials (lines 241-244); lower SES being "unrelated" to differences in beliefs about potential negative consequences of exploring (lines 266-267); SES being "unrelated" to the other model parameters (line 269). As equivalence testing requires specifying an effect size of interest prior to conducting the analyses, I do not recommend doing such analyses post-hoc. Rather, it would in my view be good (a) to change interpretations to reflect "absence of evidence", rather than implying "evidence of absence", and (b) to make explicit in the Discussion section that the null results in this paper imply "absence of evidence", not evidence of absence, because the statistical tests used do not afford conclusions about support for the null hypothesis.

6. Causal language

In my view, it's fine for the paper to use causal language when describing the theory guiding the research, but not when describing the relationships between variables in a dataset, because the data are correlational. I recommend going through the paper and adjusting any instances of the latter. To illustrate I provide two quotes. Lines 129-132: "The current study investigated whether SES shapes the balance between exploration and exploitation and, in turn, examined the impact of exploratory decisions on learning (...) and academic achievement (...)." I agree that the study examined the impact of exploratory decisions on learning (in the task), but not that it examined the impact on academic achievement (in the real-world environment). Lines 406-407: "These findings indicate that socioeconomic status can sculpt internal decision processes". I agree the data are consistent with this possibility, but would prefer a more cautious wording, because the association between SES and decision processes may have arisen by factors other than SES.

7. Cognitive processes

In my view, the BELT's (Balloon Emotional Learning Task) task structure is a double-edged sword. On the positive side, this structure mimics the structure of many important real-world problems. On the negative side, the BELT does not isolate a single cognitive process (i.e., no task purity). In particular, the BELT measures three different types of preferences: for time (sooner rather than later rewards), risk (variance in outcomes); and information (reducing uncertainty; about explosion limits). The authors have used computational modeling to uncover the psychological processes underlying decision-making—which is excellent—focusing on five parameters: loss aversion, beliefs about the balloon exploding, updating exponent, risk preference, and inverse temperature. However, there may be other psychological mechanisms at work as well, such as future discounting and impulsivity (<https://doi.org/10.1037/bul0000375>). I would not want to ask the authors to

computationally model those parameters, but if they agree these could be relevant, it might be worth acknowledging them in the Discussion section.

8. Limitations

I would appreciate a discussion of the limitations of the research. This helps readers calibrate their inferences and promotes future research addressing these limitations, fostering cumulative science. Such discussion could focus, for instance, on the BELT's psychometric properties (e.g., convergent validity with other measures of risk taking). This validity has been questioned for the related BART task (<https://doi.org/10.1037/xge0001036>). What is known (and not known) about the validity of the BELT? A brief discussion should be possible within the journal's word limit. Or discussion could focus on limitations of SES measures (e.g., self-reported, no census data).

MINOR POINTS

74: "One prominent proposes that" — the word "theory" or "hypothesis" appears to be missing before "proposes"

78: consider making the referent of "this relationship" more explicit; as it stands, "this relationship" could be interpreted as referring to the developmental explore-exploit shift occurring earlier among youth with low SES (for which the authors do discuss some evidence, and hence the word "limited" would be better suited than "lacking")

88: consider replacing "does not typically lead to" with "constrains", because 'some' new learning does often occur when exploiting a well-honed skill

89: "Relative to adults, children and adolescents are highly exploratory" — consider replacing the word "are" with "tend to be", because the evidence [that children are more exploratory than adults, and more generally that there is a linear transition from more to less exploratory with age] is more mixed than the current wording suggests (e.g., <https://doi.org/10.1098/rstb.2019.0503>)

117: consider changing the word "literature" to plural ("literatures")

121-125: "One possibility is that poorer adolescents are more loss-averse than their wealthier peers because they have fewer resources to compensate for a potential loss. This could lead to a preference to limit losses by exploiting options guaranteed to lead to favorable outcomes." — consider mentioning the 'desperation threshold' model in this context (<https://doi.org/10.1038/s41598-020-80897-8>)

216: "first task third" — consider rewording as "first third of the task", which is clearer and also used earlier

224-225: "To assess whether exploration—and the resulting explosion-related feedback—boosted performance and optimal strategies" — this wording is challenging to follow, consider slightly rewording

254-255: "individual differences in exploration are reliable" — the meaning of the word "reliable" here was initially not clear to me; please consider including the explanation provided in the Supplementary Materials—"individuals who are highly exploratory at one time point are likely to be found to be highly exploratory at another"—in the main text (and in the section titled "Individual differences in Exploration are reliable" in the data analysis script).

258: "confirmed" — consider replacing this word with "supported"

260: the term "loss sensitivity" is not defined (see also lines 282-283); also, please make explicit how it differs from the term "loss aversion" used elsewhere (defined on lines 264-265)

276-277: the meaning of the phrase "ab for pumping" is not clear to me (see also "pumping or ab" and "explosions or ab" on lines 311-312); perhaps their meaning will be obvious to others

296: "benefited" — please check spelling, because the same spelling is used line 305, but a different spelling is used in line 410 ("benefitted")

305: consider changing "particularly" to "but only" (to make the contrasting findings in the other conditions more explicit)

323-324: "Lower SES correlated with greater loss aversion but not differences in beliefs about exploration outcomes" — consider adding the word "with" before "differences"

331: please make explicit that the error bars represent one standard deviation of the mean (rather than two, shown in some papers)

456-458: "Our findings suggest that disparities in academic achievement reflect not only the distal limitations of lower-income environments but a specific proximal psychological mechanism" — consider adding the word "also" before "a specific" (to complement "not only" and emphasize that both distal and proximate factors appear to matter)

468: "Participants received compensation for their time" — consider making the type of compensation explicit (e.g., by adding the word "financial" before "compensation")

470: I wonder whether "Sensitivity analysis" refers to "power analysis" in this context

483: "4 were missing paternal/parent 2 education variables" — perhaps a comma is missing before the number "2" here

(Remarks on code availability)

I have looked at the data and script files, but not reproduced the analyses.

Version 1:

Reviewer comments:

Reviewer #1

(Remarks to the Author)

Thank you to the authors for their responses to my queries and suggestions. They have provided important clarification about their predictions, their rationale for the variables used in the analyses, and limitations to their findings. I think paper makes an important contribution towards understanding the association between explore/exploit biases and real-world outcomes, which will be of great interest to the field.

(Remarks on code availability)

I have reviewed the code, which seems sufficient to reproduce the results, though I have not had the opportunity to run this code on my own device.

Reviewer #2

(Remarks to the Author)

The authors have been highly responsive to my review. Their initially already strong manuscript has increased in clarity, precision, and comprehensiveness. The current version of the manuscript significantly advances the field -- in particular, research on exploration-exploitation, socioeconomic status, and academic attainment. I have no further suggestions, except two very minor comments:

line 466: correct spelling of the word "questionnaires"

line 469: the analyses provide no evidence linking lower SES to learning rate -- the authors may consider adding a brief interpretation of this finding (e.g., does it teach us something interesting about participants' mental models of the rate or predictability of environmental information?)

(Remarks on code availability)

I have looked at the data and script files, but not reproduced the analyses. In my view, for-profit journals can hire paid modellers to verify code and annotation, because reproducing analyses requires general computational expertise, rather than domain specific knowledge.

REVIEWER COMMENTS

Reviewer #1 (Remarks to the Author):

Thank you for the opportunity to review this manuscript. In this study, the authors examine the association between explore/exploit preferences, socioeconomic status (SES) and academic attainment in a sample of 12–14-year-olds. They find participants from lower SES backgrounds explore less than peers from higher SES backgrounds; further, lower rates of exploration were associated with lower academic attainment. This is a timely topic bridging the explore/exploit framework to real world decision-making, which thus far has been lacking in the literature. However, I feel this study has some important caveats that are not sufficiently acknowledged in the present manuscript and would need to be addressed before it is suitable for publication. Most pressingly, I think there needs to be greater acknowledgement in the text that this is cross sectional data which is an important caveat given the authors propose these findings might be used to guide policy. I have some additional comments/queries, which I outline by section below:

Thank you for your feedback and the time that you have taken to review the manuscript. We have made critical revisions to the manuscript based on your feedback. We believe that these revisions significantly strengthen the quality of the paper. Our response to each comment is detailed below.

Introduction:

1. For a more general readership, it might be helpful to expand on the summary of why exploration in childhood and adolescence is beneficial later in life. The authors cite relevant work from Gopnik and colleagues – summarizing their account in slightly greater detail would provide greater context for the importance of considering why reduced exploration may have downstream consequences in the lifespan.

Thank you for raising this important point. We have expanded on the summary about why exploration is critical in the Introduction. We now highlight that this early exploratory phase enables knowledge acquisition that allows for better problem solving and decisions in adulthood. Starting on page 3, line 94, we write:

“The human species is distinguished by an unusually extended childhood—a feature hypothesized to have evolved to enable extensive exploration before the responsibilities of adulthood take hold^{1–40}. This prolonged period is hypothesized to allow children and adolescents ample time to gather information about the world’s complex structure and optimize decision strategies to suit their early environment²³. Knowledge and strategies gained through early life exploration can be leveraged in adulthood for optimal decision-making^{21–26,30–32,34,36,39,40}. Consistent with this idea, children and adolescents tend to be more exploratory

than adults^{23–27,27,29,33,35,38,39,41}, though some studies report conflicting findings^{32,42}. For instance, children make more exploratory choices on decision tasks, which boosts learning of task structure—allowing children to better adapt to changes in task rules than adults^{27–29,40}. Adolescents also exhibit more risky exploratory behavior than adults^{22,24,25,30,33,35,38}. This is related to superior performance on foraging tasks²⁵. Based on existing frameworks^{21–40}, if this critical developmental period is missed, opportunities for broad learning would be constrained, leading to less optimal decisions later.”

2. In discussing the survival benefits of adjusting explore/exploit strategies to the availability of rewards in the environment, I suggest referring to non-human animal research that has provided stronger causal evidence for this account (see Pradhan et al. 2019; also reviewed in Lloyd et al., 2023).

Thank you for this suggestion. We now cite non-human animal research in the Introduction, including work by Pradhan et al., 2019 and Lloyd et al., 2023. These references illustrate that resource deprivation in early development has been causally linked to reduced exploration—which we believe nicely complements the correlational evidence we review in humans. On page 4, starting on line 119, we write:

“Non-human animal studies complement this cross-sectional research, showing that resource scarcity (e.g., food deprivation)^{45,49,50} causally reduces exploration later in life.”

3. The authors do not state any explicit hypotheses for the present study, though they review sufficient literature to propose directional predictions about the association between SES, exploration and academic attainment. I am not suggesting the authors should hypothesize after results are known, but it would be helpful to understand whether the authors did have specific predictions in order to understand whether the data are able to test these predictions.

The study was designed to test two *a priori* hypotheses on the relationship between SES, exploratory decisions, and learning. The first hypothesis was that lower SES would correlate with less exploration and more exploitation on the BELT. We predicted that lower SES would correlate with less exploration. The second hypothesis was inspired by research linking exploration to learning. We hypothesized that SES-based differences in task performance and academic achievement would be mediated by reduced exploration in lower-SES adolescents.

In addition to these two primary hypotheses that guided the design of the study, we added a third *a priori* hypothesis before looking at the data, but after the study was designed. We hypothesized that reduced exploration would be mediated by loss aversion and/or more pessimistic initial beliefs that exploring would lead to

negative outcomes. We now provide this information in the manuscript in the Introduction section starting on page 5, line 147:

“We tested three a priori hypotheses. First, based on prior research^{11,12,18,43–45,49,50}, we hypothesized that lower SES would correlate with less exploration. Second, we hypothesized that SES-based differences in task performance, grades, and academic skills would be mediated by reduced exploration, given established links between exploratory behavior and learning^{12,21–24,26,29,30}. Third, we hypothesized that reduced exploration in lower-SES adolescents would correlate with heightened loss aversion and/or initial beliefs that exploring would be more likely to lead to negative outcomes. The latter hypothesis was conceived before data analysis but after the study was designed.”

Methods:

4. There is important demographic information missing about the participants recruited for this study, including their ethnicity, English language ability, and any developmental disorders that could impact language skills (the latter of these points may be covered under the neurological conditions assessed, but it would be helpful to clarify). This consideration is important for several reasons:

i) Such information would be necessary to report to ensure the study is reproducible.

ii) Given evidenced SES differences responses to opt-in parental consent (Spence et al., 2015), it would be important to report demographic information to consider whether the sample was representative of low- and high-SES populations.

iii) The measures of academic attainment are highly reliant on English language ability. It would be important to understand whether any students had diagnoses that could impair their written or spoken language, which may impact measures of academic attainment.

Thank you for highlighting the need to provide more information about the sample’s demographic information. Before describing the changes we have made, we wish to note that all participants recruited into the study were proficient in English, and this is detailed as an inclusion criterion in the Methods/Participants section of the paper. In this section, we have also stated that individuals with neurological disorders and autism were not eligible to participate in the study.

We have now detailed information about ethnicity and reading/language delays in the Supplement (Supplementary Tables 1 and 2). We detail this information for the full sample, as well as separately for the higher- and lower-SES subgroups

(which we divided using a mean split on the SES data) so readers can evaluate their representativeness. These tables are pasted below for your review.

Note that the ethnicity data is comparable to that of the United States (<https://www.census.gov/quickfacts/>). The linked website details the breakdown of the United States population as 75.3% White, 13.7% Black, 6.4% Asian, 19.5% Hispanic or Latino, 1.3% Native American, and 0.3% Pacific Islander.

Supplementary Table 1. Data on Ethnicity

SES	No. of Subjects	White	Black	Asian	Hispanic	Native American	Pacific Islander	Other
Overall Sample	122	73.77	14.75	7.37	13.93	0.81	0	5.73
Higher SES	58	81.03	6.89	10.34	6.89	1.72	0	5.17
Lower SES	64 ^{&}	67.18	21.87	4.68	20.31	0	0	6.25

Values in each cell reflect a percentage of the sample. Note that some participants reported more than 1 ethnicity and therefore percentages in each row exceed 100. Subjects were divided into a higher and lower-SES subgroup using a mean split on the SES data. [&]Two participants in the lower-SES subgroup did not report ethnicity data. Thus, the number of participants that reported ethnicity data was 122 (out of 124 included in the sample). Note that these numbers are comparable to the population of the United States (<https://www.census.gov/quickfacts/>), which is comprised of 75.3% White, 13.7% Black, 6.4% Asian, 19.5% Hispanic or Latino, 1.3% Native American, and 0.3% Pacific Islander.

The number of participants who reported a language or a reading delay can be found in Supplementary Table 2, pasted below for you. As shown, 3 participants reported having a reading delay (e.g., dyslexia) and 2 participants reported having a language delay. Note that 2 participants with a reading delay also reported a language delay. Thus, the number of participants with either a reading or language delay was 3. As shown below, these participants came from the lower-SES subgroup. As a robustness check, we re-fit the mediation models focused on academic achievement after removing these participants from analyses to confirm that the pattern of results was not altered by their inclusion. These mediation analyses showed that exploratory pumping mediated SES-based differences in grades and academic skills when these participants were excluded. We are happy to report these robustness checks in the Supplementary Materials if you think it is important.

Supplementary Table 2. Reading or Language Difficulties or Delays

SES	No. of Subjects	Dyslexia or another reading difficulty or delay	Language difficulty or delay
Overall Sample	124	3*	2
Higher SES	58	0	0
Lower SES	66	3*	2

Values in each cell reflect the number of participants in the sample that reported either a reading or a language difficulty or delay. *Two participants who reported a reading difficulty or

delay also reported a language difficulty or delay. The total number of participants with either a reading or a language difficulty or delay was 3. Subjects were divided into a higher and lower-SES subgroup using a mean split on the SES data.

We now reference these tables in the Methods/Participant section. On page 20 lines 538, we write:

“Demographic information, including ethnicity, reading, and language difficulties and delays, can be found in Supplementary Tables 1 and 2.”

5. Could the authors clarify whether participants were remunerated at a single rate, or whether payment was performance-dependent? It would be helpful for the authors to provide a brief rationale for the strategy they adopted, as performance-dependent payment may increase motivation, but may also covary with SES.

We have now described that participants were remunerated at a fixed rate, regardless of task performance. This approach ensured that any potential differences in motivation for monetary rewards across SES did not drive SES-based differences in task performance. We have provided this rationale on page 20, starting on line 540:

“All participants were remunerated at a fixed rate, independent of their task performance. This ensured that any potential differences in motivation for monetary rewards across SES would not drive SES-based differences in performance. Participants received \$75 for completing both the BELT and demographic questionnaires and could earn up to \$200 for completing all aspects of the study, including MRI tasks not reported here.”

Results:

6. Greater a priori justification is needed for the choice of predictors and outcome variables in the regression and mediation analyses. The authors use pumps in the unreliable balloon trials in their analyses to test the hypothesis related to SES and exploration, but then restrict their analyses to the long pump condition when testing whether exploration mediates the association between SES and task performance. In the absence of a pre-registered analysis plan, it is difficult to understand which of these analyses were planned prior to examining the data.

Thank you for highlighting the need to clearly justify the use of specific predictors. We now detail the rationale for each decision in this response letter below and have amended the manuscript to make the rationale clearer. Below, we first justify the decision to use unreliable balloon trials as an index of individual differences in exploration throughout the paper. We then describe why we

focused on long balloon trial points in mediation models examining SES-based differences in task performance.

Using unreliable balloon trials to index individual differences in exploration

We chose *a priori* to use pumping and explosions on unreliable balloon trials to index individual differences in exploration. These metrics are used consistently throughout the paper to index individual differences. This decision was made because we anticipated that on long and short balloon trials, participants would adjust pumping behavior in response to learning the explosion points. We therefore reasoned that pumping and explosions on long and short balloon trials would be a biased measure of exploration, confounded by learning effects. As anticipated, the number of points, pumps, and/or explosions significantly changed across the task on long and short balloon trials (See Results and Analysis S1). In contrast, there was no evidence that participants learned the explosion threshold of unreliable balloon trials. We therefore reasoned that pumping and explosions on unreliable balloon trials were less biased measures of exploration and used these to index of individual differences in all analyses.

We have amended the Methods and Results to clearly justify this reasoning. In the Methods, on page 25, starting on line 687, we write:

“To assess individual differences in exploration, we used the number of pumps and explosions on unreliable balloon trials. We chose a priori to use unreliable balloon trials to index exploration because we anticipated that individuals would adjust their behavior in response to learning on the other trial types, which would confound the measure of exploration with learning effects. Consistent with this expectation, participants showed evidence of learning on the long and short balloon trials, but there was no evidence of learning on the unreliable balloon trials, making unreliable balloon trials a less biased measure of exploration (See Results and Supplementary Note 1).”

We have also justified this decision in the Results on page 9, starting on line 241:

“Next, we tested whether lower SES correlated with reduced exploration. We used pumping and explosions on unreliable balloon trials to index individual differences in exploration because pumping and explosions on the other trials—short and long balloon trials—were confounded by learning effects (See Results above and Supplementary Note 1).”

For thoroughness and to increase the credibility of the findings, we repeated the analyses relating SES to exploration, substituting the primary metric of exploration (unreliable balloon trial explosions and pumping) with pumping and explosions across all trials. The results are fully consistent with the pattern described in the primary text: Lower SES correlates with less exploration, as

shown by less pumping and fewer explosions. These findings are detailed on page 2, line 37 of the Supplement:

“In the primary manuscript, we operationalized exploration using pumps and explosions on unreliable balloon trials. We made this decision a priori because we anticipated that individuals would show evidence of learning on the long and short balloon trials, which could bias the individual difference metric of exploration. For thoroughness and to increase the credibility of the findings, we re-fit models relating SES to exploration, but substituted the primary metric of exploration with pumping and explosions on all trial types. We observed the same pattern of results as those reported in the primary paper: Lower SES correlated with less pumping ($b = 0.732$, $SE = 0.182$, $z = 4.03$, 95% CI[0.376, 1.088], $p < .001$) and fewer explosions ($b = 0.18$, $SE = 0.06$, $z = 2.83$, $p = .005$).”

Testing whether exploration mediates SES differences in task performance

The goal of the mediation analysis was to understand whether individual differences in exploration mediate SES-based differences in task performance. The measure of individual differences in exploration remained consistent in this analysis: the number of pumps and explosions on unreliable balloon trials.

Since SES-based differences in performance emerged exclusively on long balloon trials, we focused the mediation analysis on this trial type. While this focus on long balloon trials was not predetermined in our initial analysis plan, it was methodologically sound given that SES-based differences in performance were significant only for these trials. We have now justified the decision to focus on long balloon trial points in the Results. Starting page 12, line 331, we write:

“We next examined whether exploratory behavior mediated SES-related differences in task performance in the final third of the task. We focused on points on long balloon trial points because these trials drove the task performance differences across SES.”

As a robustness check, however, we have added Supplementary analyses to confirm that individual differences in exploration on unreliable balloon trials also mediate SES-based differences in task performance across all trial types (not just long balloon trials). These analyses demonstrated the same pattern of results: reduced exploration explained SES-based differences in task performance. These findings are reported in Supplementary Note 4 (page 3 starting on line 64). Thus, the finding that exploration mediated SES-based differences in task performance was robust to different operationalizations.

We refer to these analyses in the main paper (page 13, line 343): *“This pattern held when operationalizing performance or exploration using all trial types; Supplementary Note 4).”*

Finally, we re-fit the primary mediation models testing whether exploratory pumping and explosions mediate SES-based differences in grades and academic skills. However, we substituted the primary metric of exploration–pumping on unreliable balloon trials–with pumping across all trials. These analyses are reported in Supplementary Note 5 and are fully consistent with those reported in the primary paper. We refer to these analyses in the main paper on page 15, line 406.

7. Given that multiple analyses are conducted on the same dataset, I suggest correcting the analyses for multiple comparisons.

We have now applied False Discovery Rate (FDR) corrections using the Benjamini-Hochberg procedure to control for multiple comparisons. We report FDR-adjusted p-values (p-adjusted) throughout the manuscript. Analyses that were significant in the original submission remain significant after correcting for multiple comparisons.

To correct for multiple comparisons, we separated analyses into three conceptually distinct families of tests to maintain control while accounting for the independence of different hypothesis tests. The first family focuses on general learning effects, which were used to validate the task:

- The relationship between trial number and points earned
- The interaction between trial number × balloon type for predicting points earned
- The relationship between trial number and points earned separately by balloon condition

The second family focused on how exploration was related to task performance:

- The relationships between early exploration and later points
- The relationships between exploration and loss aversion
- The dynamic relationships between points and subsequent exploration

The third family of tests focused on all analyses involving SES:

- SES relationships with exploration (pumps, explosions)
- SES relationships with task performance (points)
- SES relationships with loss aversion and beliefs about the balloon exploding
- SES relationships with academic outcomes (grades, academic skills)
- All mediation effects (exploration mediation SES based differences in task performance, school grades and academic skills)

Note that we did not correct the analyses reported in the supplement for multiple comparisons because these analyses were primarily either focused on task validation, robustness checks or exploratory analyses.

We have updated the Methods section to explicitly describe the approach for dealing with multiple comparisons. On page 23-24, starting on line 640, we write:

“To control for multiple comparisons, we implemented false discovery rate (FDR) corrections separately for three conceptually distinct families of analyses reported in the main text. The first family focused on whether participants learned on the BELT, which included all analyses relating trial number to points earned. The second family of tests focused on how exploration was related to other variables including task performance (points) and loss aversion. The third family of tests included all analyses relating SES to other variables, including exploratory behavior, BELT performance, and academic outcomes. For each family, p-values were ranked from smallest to largest and compared against critical values calculated using the Benjamin-Hochberg procedure. We report both uncorrected and FDR-adjusted p-values in the text. Note that exploratory and validation analyses included in the Supplement were not corrected for multiple comparisons.”

8. The authors should be commended for the thorough analyses of the task data. However, not all of the findings presented in the results section appear to be directly related to their research question. For example, the analysis of improved learning across the task is helpful to establish the validity of the task in this age group but does not directly relate to the main research question.

Thank you for this feedback. The first two sections of the Results were used to validate the task. However, we agree that they are not directly relevant to the core research questions being asked about SES. These sections show that: (1) individuals learned on the BELT; (2) early exploration boosted later learning. We have now significantly streamlined these sections, while keeping critical validation steps, which has significantly shortened these sections.

In the first paragraph of the Results, in *“Adolescents exhibited incremental learning across trials”* (page 8, line 209), we moved all analyses on how pumping and explosions shifted across the task to the Supplement (now Supplementary Note 1). The updated section now focuses exclusively on how points increased across trials. This significantly streamlines the text.

In the second paragraph of the Results, *“Exploration Boosted Subsequent Performance on the BELT”* (page 8, line 227), we removed trial-type specific analyses focusing separately on long, short, and unreliable balloon trials. This section now only reports the relationship between early exploration and later points across all trials.

We have also removed Figure 2, which displayed panels showing how points changed across the task for each balloon type, as well as how early exploration boosted later learning.

The revised version of the first two sections of the Results is considerably shorter (404 words versus 986).

Finally, we also removed a panel from Figure 3, which displayed data showing that exploratory pumping and explosions covaried with reward outcomes—a finding that is not strictly relevant to the core research questions being asked.

These changes streamline the manuscript to spotlight our central findings about socioeconomic status while retaining essential validation steps.

9. If loss aversion is the proposed mechanism driving reduced exploration (and subsequent poorer academic attainment) in individuals from low SES backgrounds, should this parameter not also be subject to similar analyses to exploration? I.e., analysis to examine whether loss aversion mediates the association between SES and academic attainment. Such analysis should be labelled as exploratory but could provide more precise insight into how interventions may address SES disparities in education, should these findings be replicated in a longitudinal sample.

Thank you. We performed exploratory mediation analyses and found no evidence that loss aversion mediated SES-based differences in academic skills or grades. These findings suggest that loss aversion may only influence academic achievement indirectly via its impact on exploration. We have now reported these analyses in Supplementary Note 6 and labelled them as exploratory. Starting on page 4, line 117 of the Supplement, we write:

“We fit 2 exploratory mediation models to test whether loss aversion mediated SES-based differences in grades and academic skills. There was no evidence that loss aversion mediated SES-related differences in academic skills in the lower-SES subgroup (mediation/indirect effect or $ab = 0.0673$, 95% CI [-0.0838, 0.24], $p = 0.376$). Higher SES was significantly related to better academic skills (total effect or $c = 0.467$, 95% CI [1.31, 0.800], $p = 0.007$), and this relationship was not significantly reduced after accounting for loss aversion (direct effect or $c' = 0.399$, 95% CI [0.033, 0.75], $p = 0.033$). Similarly, there was no evidence that loss aversion mediated SES-based differences in grades (mediation/indirect effect or $ab = 0.020$, 95% CI [-0.042, 0.090], $p = 0.494$). Higher SES was significantly related to better grades (total effect or $c = 0.294$, 95% CI [0.119, 0.480], $p < .001$), and this relationship was not significantly reduced after accounting for loss aversion (direct effect or $c' = 0.270$, 95% CI [0.079, 0.450], $p = 0.006$). These results raise the possibility that loss aversion may influence academic achievement only indirectly via its impact on exploration.

We have pointed readers to this Supplementary Analysis in the main paper on page 15, line 406.

“Furthermore, loss aversion did not significantly mediate SES-based disparities in academic outcomes (Supplementary Note 6)—suggesting it may influence academic outcomes only indirectly via its impact on exploration.”

10. It would be helpful to provide information about the number of participants assigned to the low SES and high SES categories after the mean split was conducted.

We have now included this information along with other important demographic information about the sub-groups in the Methods and Results. In the Results section on page 14, lines 380, we write:

“To further probe how pumping related to academic skills across SES, we divided the sample into a lower- and higher-SES subsample using a mean split on the SES data. The higher-SES group comprised 58 participants (median income: \$150,000; range: \$65,000-\$1.25 million; median parental education: 17 years; range: 14-20 years). The lower-SES group comprised 66 participants (median income: \$70,000; range: \$2,000-\$200,000; median parental education: 13 years; range: 7-17 years).”

In the Methods starting on page 28, line 804, we write:

“Significant interactions were probed by re-fitting models separately for a higher- and lower-SES subgroup, divided using a mean split on the SES data (n = 58 in the higher SES group [median income: \$150,000; range: \$65,000-\$1.25 million; median parental education: 17 years; range: 14-20 years, n = 28 male, n = 30 female] and 66 in the lower SES group [median income: \$70,000; range: \$2,000-\$200,000; median parental education: 13 years; range: 7-17 years, n = 34 male, n = 32 female]).

Discussion:

11. It would be worth noting that that authors did not find evidence for improved performance in the short pump threshold condition for adolescents from low SES backgrounds. This finding is in contrast to Humphreys et al. (2016) who found early adversity conferred benefits in low threshold conditions. While the operationalization of early life experience (adversity versus socioeconomic status) is different between the studies, it would be helpful to frame the findings in context of this previous research if the authors are to consider the implications for educational settings.

Thank you. We have integrated the findings from Humphreys et al. (2015) *Exploration-exploitation strategy is dependent on early experience*, into the Discussion. In this paper, Humphreys et al. (2015) showed that a history of institutionalization was associated with less exploration across all conditions, including the restricted condition. However, this reduced exploration did not translate to more points earned in the restricted condition. This finding is actually consistent with the findings reported in the present paper, where lower SES was associated with less exploration across conditions, but not to performance advantages in the short balloon condition that rewarded an exploitative strategy. We have now highlighted these interesting patterns and provided a potential explanation on page 17, starting on line 446:

“Although heightened exploitation could theoretically confer advantages in certain contexts, lower-SES adolescents did not display performance advantages over their affluent peers in the task condition that rewarded exploitation. This result may seem surprising, but it aligns with findings showing that early adversity (i.e., institutionalization) increases exploitation, without leading to performance advantages in a condition that favors exploitation⁴³. One explanation for this apparent paradox in the present study is that higher-SES adolescents learned the optimal strategy for each condition and adapted their behavior accordingly. Their flexibility in strategy use would have cancelled out performance advantages that lower-SES adolescents could have gained from generalizing an exploitative approach across conditions. Thus, addressing SES-based disparities in performance may require supporting context-appropriate decision strategies that boost exploration but only when it is optimal.”

12. The authors should discuss limitations to the present study. Most importantly, it would be necessary to highlight that the cross-sectional data precludes any causal inference being drawn from this study and I think a greater recognition of this limitation throughout the discussion (e.g., in the summary of the results in the first paragraph of the discussion) would provide a more accurate summary of the contribution of this study to the literature. Discussing the limitations of the design would also be important when reflecting on the choice of analyses, as mediation models are not always appropriate for cross-sectional data (see Cain et al., 2018). At the very least, this limitation should be discussed with relation to the conclusions that can be drawn from these data and the type of analysis used.

We have now stated that the data is cross-sectional, which limits the ability to make causal inferences in several places in the Discussion.

In the first paragraph of the Discussion, starting on page 17 line 434, we write:

“While the cross-sectional design limits causal inference, these findings align with theoretical frameworks that early SES shapes exploratory decisions, which may, in turn, limit learning and achievement.”

Furthermore, in this same paragraph, we call for future research that can establish causal relationships. On page 17, starting on line 443, we write:

“These exploratory decisions correlate with learning and academic achievement, though future studies will be essential to establish causal relationships.”

We have also added a limitations paragraph to the Discussion, where we highlight the cross-sectional nature of the study. Starting on page 19, line 511:

“Finally, the cross-sectional design and mediation analyses preclude the establishment of causal inferences⁷⁰. It is our hope that these correlative findings inspire future studies assessing the impact of SES on decision strategies and learning.”

Finally, we have gone through the manuscript and removed instances where we had used causal language such as “influences” and “shapes” and replaced it with more appropriate descriptions.

Minor:

p.3 line 74 missing the word ‘theories’ after ‘prominent’

Thank you. We have corrected this.

References:

Cain, M. K., Zhang, Z., & Bergeman, C. S. (2018). Time and other considerations in mediation design. Educational and psychological measurement, 78(6), 952-972.

Lloyd, A., Viding, E., McKay, R., & Furl, N. (2023). Understanding patch foraging strategies across development. Trends in Cognitive Sciences.

Pradhan, S., Quilez, S., Homer, K., & Hendricks, M. (2019). Environmental programming of adult foraging behavior in *C. elegans*. Current Biology, 29(17), 2867-2879.

Spence, S., White, M., Adamson, A. J., & Matthews, J. N. (2015). Does the use of passive or active consent affect consent or completion rates, or dietary data quality? Repeat cross-sectional survey among school children aged 11–12 years. BMJ open, 5(1), e006457.

Reviewer #1 (Remarks on code availability):

Given my reservations about the study in its current form, I have opted not to

review the code on this occasion and hope this is okay. If the authors are invited to revise and resubmit the paper, I would be very happy to review the code to ensure it is computationally reproducible.

Reviewer #2 (Remarks to the Author):

This well-written paper is theoretically interesting and timely, and will likely be of interest to the readership of Nat Comm. It certainly has the potential to make a valuable contribution. However, the paper could be improved along several dimensions. Below I list 8 major points, all of which can be resolved, and several minor points. I would look forward to reading a revised version of this paper.

Thank you. We appreciate your constructive feedback and the time that you have taken to review the manuscript. We have responded to each of your comments below.

MAJOR POINTS

1. Data and script. I appreciate the authors sharing their data and script. However, in the data file, I was unable to find descriptions of the variables (only labels); and the script included minimal annotation. To ensure reproducibility, it would be good to improve these aspects of documentation.

We have gone through the script and provided line-by-line annotations throughout. To connect the code with the manuscript, we have also added relevant result headings above each code section to show where specific analytic output appears in the paper. Additionally, we created several README files that provide complete information about the datasets, including variables, the purpose of each script, and step-by-step instructions on how to re-fit models.

2. Preregistration. The paper reports confirmatory research (line 623); that is, it uses data to test hypotheses, rather than to generate hypotheses (as done in exploratory research). However, I could not find a mention of preregistration (<https://doi.org/10.1073/pnas.1708274114>). If the research was not preregistered, it would be good to make this explicit (a) when introducing the current study (e.g., on line 135, the authors could add: “The current research was not preregistered.”) and (b) to mention it as a limitation in the Discussion section.

We now state explicitly on page 5 line 156 of the Introduction:

“While this study was hypothesis-driven, it was not pre-registered. All data and code can be found on OSF: <https://osf.io/hwt6z/>.”

In the Discussion, we have also detailed several limitations of the work, including the lack of pre-registration. On page 19, starting on line 499, we write:

“The present study has several important limitations that could be addressed in future research. First, the study was not pre-registered, a step that enhances transparency and reproducibility in scientific research.”

3. Methods and participants. The information on methods and participants seems to be incomplete (and consistent in one place). Please specify the place(s) from which the sample was drawn (e.g., country, region, or city). It would be good to include this information in the abstract and in the “Participants” section in the main text. Please specify the test setting (e.g., online, lab, community center). Third, the number of participants included in the analyses of grades is initially described as 114 (line 393) and later as 115 (lines 521-522). Fourth, the upper end of the range of household income is initially described as “1.1 million” (line 137) and later as “\$1.25 million” (line 476). Fifth, it would be valuable to provide more information about the distributions of both the component and the composite SES measures (in a table or figure in the Supplementary Materials).

Thank you for catching this missing information and inconsistencies. This has helped us to more thoroughly and accurately report the study procedures and demographic information about participants, improving the quality of the manuscript.

First, in line with your feedback, we now state in the abstract that participants were from the United States:

“Here, we show that lower socioeconomic status is associated with less exploration on a reward learning task (n=124, 12-14-year-olds from the United States).”

Furthermore, the Methods/Participant states on page 20 line 527 that participants were recruited through flyers, social media, and local schools *“in and around the Boston area”*.

We have also indicated that the study took place in a quiet testing room at MIT, where participants completed the BELT on a laptop computer. On page 290-21, starting on line 556, we write:

“All tasks were completed in a quiet testing room. Participants completed the BELT on a laptop computer and the Woodcock-Johnson Test of Achievement (Edition IV) was administered by trained research staff.”

Fourth, we have clarified the number of participants included in analyses focused on grades and academic skills. During our review of the data, we discovered that

we had accessed an incomplete record for the academic achievement measures, with several participants tested later in the data collection period having missing values. Upon accessing the complete records, we found that we had grades for 122 of 124 participants (only 2 missing) and academic skills for all 124 participants. We have rerun analyses using the complete dataset. The results remain unchanged. SES shows robust associations with both grades and academic skills (p s < 0.01). Furthermore, all mediation analyses are consistent with what was reported initially, showing that exploratory pumping mediates SES-based differences in academic skills and grades, with confidence intervals that do not include 0. All figures and tables now reflect the correct sample sizes.

Fifth, we have corrected the inconsistency in the reporting of the upper bound for household income, which is 1.25 million.

Finally, we have added Figures 1a and b in the main paper to display the distribution of household income and parental education for readers. We have also added a Supplementary figure (Figure S1) to show the distribution of the composite SES score derived from parental education and income. These figures are pasted below for your review.

Figure 1. Annual Household Income and Parental Education Distributions, and the Balloon Emotional Learning Task Schematic. a) The annual household income distribution of the sample plotted on a log scale ($n=121$; median = \$100,000; range: \$2000-\$1.25 million). The dashed line marks the low-income threshold for a family of four in 2019 in the Boston-Cambridge-Quincy Metro Area (\$89,200), and the brown solid line marks the median household income in 2019 in the Boston-Cambridge-Quincy Metro Area (\$113,300); retrieved from: https://www.huduser.gov/portal/datasets/il.html#2019_query. b) The distribution of parental years of education separately for maternal/parent 1 (median: 16 years; range: 7-20; $n=124$) and paternal/parent 2 education levels (median: 14 years; range: 7-20; $n=119$). In A-B, Y-axes reflect density distributions as percentages.

Supplementary Figure 1. Composite SES score, reflecting the mean of the z-score for parental education and log income (n=124). For all panels, Y-axis reflects percentages.

4. Calibrated conclusions. There is scope to better calibrate conclusions to the data. The authors compared two measures of exploration (number of pumps and explosions) with two measures of academic achievement (grades and academic skills). The data support associations between pumping and both measures of academic achievement, but do not support associations between explosions and either measure of academic achievement. This is described on lines 356-362: “more pumping on the BELT correlated with better academic skills (...), although the mean number of explosions were unrelated to academic skills (...). More pumping (...), although not more explosions (...) on unreliable balloon trials predicted higher grades.” Despite the evidence being mixed, the authors proceed as though the evidence provides consistent support. For instance, the quoted paragraph concludes with: “Thus, greater exploratory tendencies correlated with better academic achievement.” In my view, this conclusion would be better calibrated if it read: “Thus, greater exploratory tendencies correlated with better academic achievement on some measures, but not others”. Similar statements occur later in the paper, for instance, on lines 445-445: “we observed that reduced exploratory tendencies were related to lower academic achievement”. The paper could do more justice to the mixed nature of the evidence.

We have amended the manuscript to better calibrate the interpretation and conclusions. In the Results section, after reporting the relationships between exploration and academic achievement, we write:

“Thus, greater exploration significantly correlated with better academic achievement for one measure of exploration but not the other.” (page 14 starting on line 372).

Furthermore, in the Discussion, on page 18, starting on line 480, we now write:

“Consistent with this idea, exploratory pumping on the BELT statistically mediated SES-based disparities in grades, and in a lower-SES subsample, academic skills.”

5. Interpretation of null results.

196-198: “we used unreliable balloon trials to quantify individual differences in exploration, as these trials were immune to learning related adjustments behavior” — strictly speaking, the analyses support the conclusion that there is no evidence for learning (absence of evidence), but not the conclusion that there is evidence of no learning (evidence of absence). To establish the latter, one would need to conduct analyses that can test support “for” the null hypothesis (rather than only “against” it), such as equivalence testing (<https://doi.org/10.1177/2515245918770963>). The same issue applies to the discussions of exploration not boosting performance on the short balloon trials (lines 241-244); lower SES being “unrelated” to differences in beliefs about potential negative consequences of exploring (lines 266-267); SES being “unrelated” to the other model parameters (line 269). As equivalence testing requires specifying an effect size of interest prior to conducting the analyses, I do not recommend doing such analyses post-hoc. Rather, it would in my view be good (a) to change interpretations to reflect “absence of evidence”, rather than implying “evidence of absence”, and (b) to make explicit in the Discussion section that the null results in this paper imply “absence of evidence”, not evidence of absence, because the statistical tests used do not afford conclusions about support for the null hypothesis.

We have edited the Results and Discussion sections so that our conclusions and interpretations more appropriately reflect the types of statistical tests used. We now highlight that null effects reflect an absence of evidence rather than evidence of absence. We have gone through the entire manuscript to ensure these changes are reflected throughout and give some examples below.

From the Results section:

On page 9, starting on line 259, we write: *“In contrast, there was no statistically significant association between SES and initial beliefs about the probability of the balloon exploding.”*

On page 9, line 263, we write *“Similarly, there was no statistically significant association between SES and updating in response to recent outcomes, risk preference, and decision consistency”*

From the Discussion section:

On page 18 line 466, we write: *“There was no evidence that SES was related to the parameter capturing initial beliefs about the probability of the balloon exploding.”*

Finally, we have made it explicit in the limitations paragraph of the Discussion that the null results in this paper imply “absence of evidence”, rather than “evidence of absence”. On page 19, line 509, we write:

“Third, the null effects demonstrate the absence of evidence rather than evidence of absence, as our statistical tests cannot provide direct support for null hypotheses without pre-specified equivalence testing.”

6. Causal language. In my view, it’s fine for the paper to use causal language when describing the theory guiding the research, but not when describing the relationships between variables in a dataset, because the data are correlational. I recommend going through the paper and adjusting any instances of the latter. To illustrate I provide two quotes. Lines 129-132: “The current study investigated whether SES shapes the balance between exploration and exploitation and, in turn, examined the impact of exploratory decisions on learning (...) and academic achievement (...).” I agree that the study examined the impact of exploratory decisions on learning (in the task), but not that it examined the impact on academic achievement (in the real-world environment). Lines 406-407: “These findings indicate that socioeconomic status can sculpt internal decision processes”. I agree the data are consistent with this possibility, but would prefer a more cautious wording, because the association between SES and decision processes may have arisen by factors other than SES.

We have edited multiple sentences in the Introduction, Results, and Discussion to eliminate words like “shapes”, “influences”, “sculpts” and “impacts”, replacing them with words like “related to” and “correlates with”. We also highlight that the data are “consistent” with theories that SES influences exploration and learning, but that we cannot draw causal conclusions due to the cross-sectional data. We only use causal language when describing theories/frameworks, and never when interpreting data. We have provided several examples of these changes.

Starting on page 5, line 144, we now write:

*“The current study examined the *association* between SES, exploratory behavior, task performance, and academic achievement, as measured by grades and academic skills.”*

In the first paragraph of the discussion, starting on page 17, line 429, we write:

*“The present study examined the *relationship* between SES, task-based exploration, and real-world academic achievement”*

On page 17, line 434, we write:

“While the cross-sectional design limits causal inference, these findings align with theoretical frameworks that early SES shapes exploratory decision making, which in turn, limits academic learning and achievement.

We also added a paragraph to the Discussion to highlight the limitations of the study, including the cross-sectional design. On page 19, starting on line 512, we write:

“Finally, the cross-sectional design and mediation analyses performed preclude the establishment of causal inferences⁶⁷.”

We believe that these changes provide more calibrated conclusions based on the cross-sectional design of the study.

7. Cognitive processes. In my view, the BELT’s (Balloon Emotional Learning Task) task structure is a double-edged sword. On the positive side, this structure mimics the structure of many important real-world problems. On the negative side, the BELT does not isolate a single cognitive process (i.e., no task purity). In particular, the BELT measures three different types of preferences: for time (sooner rather than later rewards), risk (variance in outcomes); and information (reducing uncertainty; about explosion limits). The authors have used computational modeling to uncover the psychological processes underlying decision-making—which is excellent—focusing on five parameters: loss aversion, beliefs about the balloon exploding, updating exponent, risk preference, and inverse temperature. However, there may be other psychological mechanisms at work as well, such as future discounting and impulsivity (<https://doi.org/10.1037/bul0000375>). I would not want to ask the authors to computationally model those parameters, but if they agree these could be relevant, it might be worth acknowledging them in the Discussion section.

Thank you. We have now added a paragraph to the Discussion detailing how other underlying cognitive processes, including future discounting could contribute to disparities in decision making across SES. On page 18, starting on line 470, we write:

“However, other cognitive processes not modelled here could have contributed. For instance, lower SES correlates with steeper temporal discounting^{14,16,17,58,59}—a preference for earlier rewards—which might have led to less exploratory pumping to gain points sooner. Future work incorporating temporal discounting into a formal computational model could help test this mechanism.

8. Limitations. I would appreciate a discussion of the limitations of the research. This helps readers calibrate their inferences and promotes future research addressing these limitations, fostering cumulative science. Such discussion could focus, for instance, on the BELT’s psychometric properties (e.g.,

convergent validity with other measures of risk taking). This validity has been questioned for the related BART task (<https://doi.org/10.1037/xge0001036>). What is known (and not known) about the validity of the BELT? A brief discussion should be possible within the journal's word limit. Or discussion could focus on limitations of SES measures (e.g., self-reported, no census data).

We have added a limitations paragraph to the Discussion (page 19, line 499):

“The present study has several important limitations that could be addressed in future research. First, the study was not pre-registered, a step that enhances transparency and reproducibility in scientific research⁶⁴. Second, while the BELT demonstrates strong test-retest reliability and moderate correlations with personality traits linked to risk-taking⁵⁴, its relationship to real-world exploration remains untested. However, the Balloon Analogue Risk Task (BART), from which the BELT was adapted, shows moderate correlations with self-reported risk-taking behaviors in adolescents⁶⁵ and adults^{66,67}, though the strength of these associations varies across different measures of risk-taking⁶⁷. Given its derivation from the BART, the BELT may similarly predict real-world exploratory and risk-taking behaviors. However, this hypothesis requires direct empirical validation. Third, our analyses of null effects demonstrate absence of evidence rather than evidence of absence, as our statistical tests cannot provide direct support for null hypotheses without pre-specified equivalence testing. Finally, the cross-sectional design and mediation analyses preclude the establishment of causal inferences⁶⁸. It is our hope that these correlative findings inspire future longitudinal and intervention studies assessing the impact of SES on decision strategies and learning.”

MINOR POINTS

74: “One prominent proposes that” — the word “theory” or “hypothesis” appears to be missing before “proposes”

Thank you. We have corrected this.

78: consider making the referent of “this relationship” more explicit; as it stands, “this relationship” could be interpreted as referring to the developmental explore-exploit shift occurring earlier among youth with low SES (for which the authors do discuss some evidence, and hence the word “limited” would be better suited than “lacking”)

Thank you. We have now clarified that we are referring to the relationships between SES, exploratory behavior, and learning.

88: consider replacing “does not typically lead to” with “constrains”, because ‘some’ new learning does often occur when exploiting a well-honed skill

We have used the word “constrains” as you have suggested.

89: “Relative to adults, children and adolescents are highly exploratory” — consider replacing the word “are” with “tend to be”, because the evidence [that children are more exploratory than adults, and more generally that there is a linear transition from more to less exploratory with age] is more mixed than the current wording suggests (e.g., <https://doi.org/10.1098/rstb.2019.0503>)

Thank you. We now use the words “tend to be” and have cited the Pelz & Kidd (2020) and Somerville et al., (2017) papers showing conflicting findings.

Consistent with this idea, children and adolescents tend to be more exploratory than adults^{23-27,27,29,33,35,38,39,41}, though some studies report conflicting findings^{32,42}. “

117: consider changing the word “literature” to plural (“literatures”)

We have changed literature to literatures.

121-125: “One possibility is that poorer adolescents are more loss-averse than their wealthier peers because they have fewer resources to compensate for a potential loss. This could lead to a preference to limit losses by exploiting options guaranteed to lead to favorable outcomes.” — consider mentioning the ‘desperation threshold’ model in this context (<https://doi.org/10.1038/s41598-020-80897-8>)

We have now highlighted that individuals near a desperation threshold may actually exhibit elevated risk taking because risk taking offers the potential to escape their desperate situation. The paper suggests that for individuals near this threshold, the potential benefits of risky behavior (a large pay off) outweigh the costs (e.g., punishment if caught) as their situation cannot worsen significantly if they fail. Starting on line 137, page 5, we write:

“One explanation may be that lower-SES adolescents are more loss-averse than their higher-SES peers; they weigh potential losses more heavily than equivalent gains. This heightened loss aversion could stem from having too few resources to compensate for a potential loss (though, notably, this pattern may reverse in cases of extreme poverty, where individuals become more risk-seeking due to having so few resources left to lose and much to gain⁵¹).”

216: “first task third” — consider rewording as “first third of the task”, which is clearer and also used earlier

We have made this change.

224-225: “To assess whether exploration—and the resulting explosion-related feedback–boosted performance and optimal strategies” — this wording is challenging to follow, consider slightly rewording

This sentence now reads:

“To assess whether exploratory pumping and explosions boosted performance, we tested whether more pumps and explosions early in the task preceded more points earned later.”

254-255: “individual differences in exploration are reliable” — the meaning of the word “reliable” here was initially not clear to me; please consider including the explanation provided in the Supplementary Materials—“individuals who are highly exploratory at one time point are likely to be found to be highly exploratory at another”—in the main text (and in the section titled “Individual differences in Exploration are reliable” in the data analysis script).

We now describe what we mean by reliability in the manuscript on page 25 starting on line 695:

“Supplementary Note 7 shows that individual differences in exploratory pumping and explosions were reliable within participants, meaning those who were highly exploratory at some time points were highly exploratory at others.”

258: “confirmed” — consider replacing this word with “supported”

Thank you. We have changed this wording.

260: the term “loss sensitivity” is not defined (see also lines 282-283); also, please make explicit how it differs from the term “loss aversion” used elsewhere (defined on lines 264-265)

We apologize for the inconsistency in wording. We have removed references of “loss sensitivity” and now consistently refer to “loss aversion”.

276-277: the meaning of the phrase “ab for pumping” is not clear to me (see also “pumping or ab” and “explosions or ab” on lines 311-312); perhaps their meaning will be obvious to others.

We have now more clearly stated throughout the paper that this refers to the indirect/mediation effect.

296: “benefited” — please check spelling, because the same spelling is used line 305, but a different spelling is used in line 410 (“benefitted”)

We now consistently use consistent spelling, “benefited”.

305: consider changing “particularly” to “but only” (to make the contrasting findings in the other conditions more explicit)

We edited this sentence so as not to imply there was no effect (i.e., to refrain from interpreting a null effect) to align with the requirements at Nature Communications, which explicitly state authors cannot interpret a null effect. This sentence now reads:

“Thus, higher-SES adolescents displayed distinct learning advantages in the condition in which exploration was beneficial.”

323-324: “Lower SES correlated with greater loss aversion but not differences in beliefs about exploration outcomes” — consider adding the word “with” before “differences”

We have amended this sentence.

331: please make explicit that the error bars represent one standard deviation of the mean (rather than two, shown in some papers)

We have now removed Figure 2, which showed error bars, from the paper. All figures now explicitly state that the grey shading reflects 95% confidence intervals around the line of best fit.

456-458: “Our findings suggest that disparities in academic achievement reflect not only the distal limitations of lower-income environments but a specific proximal psychological mechanism” — consider adding the word “also” before “a specific” (to complement “not only” and emphasize that both distal and proximate factors appear to matter)

We have added the word “also”.

468: “Participants received compensation for their time” — consider making the type of compensation explicit (e.g., by adding the word “financial” before “compensation”)

We have now provided more details about participants’ compensation. We write:

“All participants were remunerated at a fixed rate, independent of their task performance. This ensured that any potential differences in motivation for

monetary rewards across SES would not drive SES-based differences in task performance. Participants received \$75 for completing both the BELT and demographic questionnaires and could earn up to \$200 for completing all aspects of the study, including MRI tasks not reported here.”

470: I wonder whether “Sensitivity analysis” refers to “power analysis” in this context

We have corrected this in the manuscript.

483: “4 were missing paternal/parent 2 education variables” — perhaps a comma is missing before the number “2” here

Thank you for pointing this out. We have clarified the phrasing.

Reviewer #2 (Remarks on code availability):

I have looked at the data and script files, but not reproduced the analyses.

REVIEWERS' COMMENTS

Reviewer #1 (Remarks to the Author):

Thank you to the authors for their responses to my queries and suggestions. They have provided important clarification about their predictions, their rationale for the variables used in the analyses, and limitations to their findings. I think paper makes an important contribution towards understanding the association between explore/exploit biases and real-world outcomes, which will be of great interest to the field.

Thank you for the time and effort that you took to review the manuscript! We feel that your feedback strengthened the paper and are glad to hear that you think it makes an important contribution to the field.

Reviewer #1 (Remarks on code availability):

I have reviewed the code, which seems sufficient to reproduce the results, though I have not had the opportunity to run this code on my own device.

Reviewer #2 (Remarks to the Author):

The authors have been highly responsive to my review. Their initially already strong manuscript has increased in clarity, precision, and comprehensiveness. The current version of the manuscript significantly advances the field -- in particular, research on exploration-exploitation, socioeconomic status, and academic attainment. I have no further suggestions, except two very minor comments:

Thank you for reviewing the manuscript! We feel that your feedback strengthened the credibility and clarity of the paper and we are glad that you believe that the paper advances the field.

line 466: correct spelling of the word "questionnaires"

Thank you! We have corrected this typo.

line 469: the analyses provide no evidence linking lower SES to learning rate -- the authors may consider adding a brief interpretation of this finding (e.g., does it teach us

something interesting about participants' mental models of the rate or predictability of environmental information?)

Thank you for this thoughtful suggestion! We agree that exploring the association between SES and learning rate could be interesting. However, in accordance with *Nature Communications'* editorial guidelines, we have refrained from interpreting null results based on frequentist statistics without additional analyses (e.g., equivalence testing or Bayesian approaches). To remain compliant with these guidelines, we have not added further interpretation of this null finding to the Discussion. We appreciate the reviewer's perspective and would be interested in pursuing this question in future work using appropriate statistical methods.

Reviewer #2 (Remarks on code availability):

I have looked at the data and script files, but not reproduced the analyses. In my view, for-profit journals can hire paid modellers to verify code and annotation, because reproducing analyses requires general computational expertise, rather than domain specific knowledge.